# Retrospective Adversarial Replay for Continual Learning

**Lilly Kumari**
University of Washington
`lkumari@uw.edu`

**Shengjie Wang**
ByteDance
`shengjie.wang@bytedance.com`

**Tianyi Zhou**
University of Maryland
`zhou@umiacs.umd.edu`

**Jeff Bilmes**
University of Washington
`bilmes@uw.edu`

## Abstract

Continual learning is an emerging research challenge in machine learning that addresses the problem where models quickly fit the most recently trained-on data but suffer from catastrophic forgetting of previous data due to distribution shifts — it does this by maintaining a small historical replay buffer in replay-based methods. To avoid these problems, this paper proposes a method, "Retrospective Adversarial Replay (RAR)", that synthesizes adversarial samples near the forgetting boundary. RAR perturbs a buffered sample towards its nearest neighbor drawn from the current task in a latent representation space. By replaying such samples, we are able to refine the boundary between previous and current tasks, hence combating forgetting and reducing bias towards the current task. To mitigate the severity of a small replay buffer, we develop a novel MixUp-based strategy to increase replay variation by replaying mixed augmentations. Combined with RAR, this achieves a holistic framework that helps to alleviate catastrophic forgetting. We show that this excels on broadly-used benchmarks and outperforms other continual learning baselines especially when only a small buffer is available. We conduct a thorough ablation study over each key component as well as a hyperparameter sensitivity analysis to demonstrate the effectiveness and robustness of RAR.

## 1 Introduction

Traditional supervised machine learning methods often rely on the assumption that the data is drawn i.i.d. from a stationary probability distribution. This assumption does not hold in many practical scenarios where the learner must continuously learn online and adapt to new tasks without revisiting previous tasks (and those tasks' data). This has motivated research in Continual Learning (also referred to as Lifelong Learning and Incremental Learning), where a machine learning model learns from a stream of data coming from a succession of different tasks [29, 45, 66].

The primary challenge in continual learning (CL) is to alleviate the "catastrophic forgetting" of the previously learnt tasks after learning new tasks [18, 39, 49]. This is mainly caused by a shift in the distribution of inputs and labels over time. For example, as a model is updated using new-task gradients, the hidden-layer representations, encoding information about previous tasks, become biased towards these new tasks. This leads to a model confusing the former with current tasks, thus producing incorrect predictions on older tasks. A widely studied strategy to alleviate forgetting is experience replay (ER) [10, 49, 50], which repeatedly trains the model on buffered replay data from previous tasks while learning the current task. However, the buffer in practice is often quite small [63], e.g., in autonomous driving, where the incremental data is large-scale and essentially never-ending. A model

36th Conference on Neural Information Processing Systems (NeurIPS 2022).

can thus over-fit to such small buffered data. Although forgetting is mitigated, the goal of retaining broad and accurate knowledge about previous tasks is not achieved, and the problem remains severe.

An ideal strategy to address the above challenge is to focus on the replay of "marginal samples" easier to forget and confuse with the current task's data, e.g., those near the boundary between the previous tasks' data and the current task's data. However, selecting such samples [2, 55] from a small buffer only brings limited improvement because the buffer (often formed using reservoir sampling [62] and thus is uniform) does not well cover many such marginal samples. Hence, we surmise that the core challenge is how to generate a small and sufficient set of samples, ones that are easier to forget and hence be confused with current task's data — given the limited buffer size, it is also important for these samples to be diverse and thus efficient.

In this paper, we propose a novel targeted adversarial synthesis based replay method, Retrospective Adversarial Replay (RAR), that can generate more informative replay instances with richer variations optimized to capture the forgetting frontier of continual learning, from a limited buffer. Specifically, given an incoming batch of the current task's data, RAR first identifies the most-likely-to-be-forgotten buffer samples (Stage I in Fig. 1) and pairs them with their nearest neighbors within the current task's data (Stage II in Fig. 1). For each pair, we then apply a bounded and targeted adversarial perturbation to the buffer sample, which moves the buffer sample's hidden-layer representation towards that of the paired current task's sample (Stage III in Fig. 1). This three-step procedure, as illustrated in Fig. 1, generates fine-grained adversarial augmentations of the buffered data that capture "forgetting" at the local boundary to the current task's data. We then effectively reduce the risk of forgetting by replaying these perturbed samples. To improve the diversity and variation of the perturbed samples (thus increasing the learnable information from a limited-size buffer), we investigate the role of MixUp [58, 67, 68], a data augmentation technique applied together with RAR — we find that it brings substantial improvements when there are strict buffer size constraints.

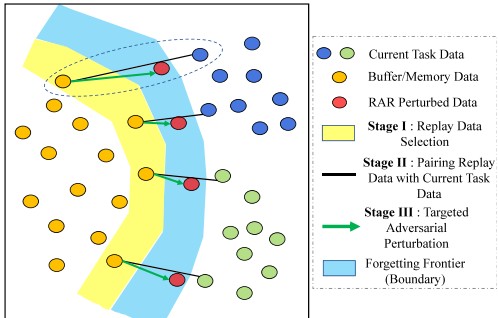

Figure 1: Our proposed RAR framework

For Stage I replay data selections, we can utilize any existing methods [2, 3, 10, 55, 65] resulting in an effective and data-efficient replay solution in CL, hence making RAR generic and complementary to existing memory-retrieval & update methods. Extensive experiments on RAR and comparisons to SoTA baselines from different CL categories demonstrate the advantages of RAR in mitigating catastrophic forgetting and improving the overall accuracy. Moreover, we conduct an ablation study showing that the key components such as replay samples selection strategy, sample pairing & adversarial perturbation, MixUp, etc, each bring appreciable improvements. Additional sensitivity analysis of the key hyperparameters implies that RAR is stable and consistently achieves these improvements.

**Connection to human false memories & their correction**: In human psychology, "false memories" [36, 42] refer to a wide variety of human memory errors ranging from the minor misremembering of small details (e.g., distorted recollection) all the way to unmitigated fabrication. The passage of time after experiencing an event increases the likelihood of these phenomena. They are referred to as [42] a "natural by-product of a distributed memory system (DMS)". Rather than a simple search & recollect, DMS [12] helps us in recombining and reconstructing memories via the integration of episodes from the past and present, but it is not always error-free. It is also true that when high-confidence false memories are corrected, people attend more to the provided feedback [17]. Despite a tendency for belief perseverance [4], the information that breaks through this perseverance is greatly attended to. The motivation behind RAR is in line with this theory. In RAR, we first combine/pair the replay buffer samples from previously seen tasks with the current data. Then using the pairing, we find existing high-confidence false memories in the form of targeted adversarial perturbations that move the previous data closer to the current data in the model's latent space. Finally, we train the model to produce the correct target for the perturbed input during the replay stage. On the contrary, existing ER-based CL methods lack this adversarial interpolation between past and present data but mainly rely on simple retrieve and replay of past data.

## 2   Related Work

Existing approaches in CL can be broadly characterized into two groups based on the model's architecture. Methods dealing with *dynamic architecture* maintain separate parameters for each task [34, 53, 64]. These methods require the knowledge of task descriptors, are expensive to train and often do not scale well when dealing with a long sequence of online tasks. In this work, we focus on the *task-free setting of CL* [61]. *Fixed architecture* approaches use *regularization* techniques to selectively regularize parameters (either weights or outputs) important for old tasks [1, 8, 27, 32, 35, 52, 66], and require task information in general. This category also includes *memory-based* approaches for constrained parameter updates on new samples to minimize interference with previous tasks [9, 37] and for replaying them along with the new task's data [2, 3, 5, 6, 10, 13, 38, 41, 43, 44, 47, 48, 59, 55, 65]. Other ER methods [2, 51, 56, 60] train *generative models* on previous tasks which can be challenging owing to the online (single-pass) learning setting.

**CL using Adversarial Information**: ACL [16], belonging to the *dynamic architecture* category, uses adversarial learning to train a shared feature encoder and a discriminator using a GAN's minimax objective [19]. ASER [55] proposes using the kNN-specific Shapley value [24, 54] to select replay samples that are representative of the buffer and adversarially close to the decision boundaries of incoming classes. GMED [25] uses gradient updates to edit buffered samples individually based on their loss increase after a look-ahead update performed only on the incoming batch. Compared to these methods, RAR is a *fixed architecture task-free CL* framework that leverages the local pairwise relationships between the replay and current task's samples to locally perturb each replay sample in a targeted adversarial manner. Our targeted perturbation generates samples close to the forgetting frontier that are hence likely confusing for the model but that are also representative of previous data — this results in a good set to further train on. To elucidate the differences in detail, we also provide a more extensive survey of the relevant prior literature in Appendix B.

## 3   Retrospective Adversarial Replay

Formally, the continual learning problem consists of a stream of data distributions $(\mathcal{D}_1, \mathcal{D}_2, \ldots, \mathcal{D}_T)$ that corresponds to $T$ tasks (in this work, the boundaries between tasks are not known). We get data points uniformly sampled from the data distribution of the current task. Our target is to train a machine learning model $f(\cdot; \theta)$ parameterized by $\theta$ that minimizes the losses as we learn each task in the given sequential order without increasing the loss on previously learned samples. In other words, the objective is

$$\min_{\theta} \sum_{i=1:T} \mathbb{E}_{(x,y) \sim \mathcal{D}_i} \ell(x, y; \theta) \tag{1}$$

Here, $\ell$ denotes the loss function and $i$ is an index over the set of tasks. To retain the knowledge of previous tasks and mitigate catastrophic forgetting, similar to [2, 55, 65], we utilize a limited memory buffer to store samples representing previously seen data. Let $\mathcal{M}$ denote the memory/buffer, which is essentially a set of data points and $|\mathcal{M}| \leq m$. Similar to ER (Alg. 2), as we train on task $i$, we get an incoming batch $(X_i, Y_i) \sim D_i$, and we combine it with a subset of memory samples $(X'_M, Y'_M) \subset \mathcal{M}$ to compute the overall loss. We can then update the memory $\mathcal{M}$ by replacing some samples with the ones encountered in the current training step while keeping the size of the buffer unchanged.

In general, ER-based methods typically deal with two problems: (1) *Memory Update* for populating the buffer and (2) *Memory Retrieval* for selecting samples from the buffer for replay. In this work, we focus on synthesizing adversarial memory samples which provide the most conflicting information on the look-ahead probing step. In other words, the model trained only on the incoming task (one-step look-ahead) misclassifies the synthesized samples (anchored around previously seen classes stored in the memory) as the incoming classes. Thus, by learning on such adversarially synthesized memory samples in retrospect, we prevent any conflicts in the probing step. For problem (1), we use reservoir sampling to update the memory (Appendix C), which can give us uniform samples from the streaming data points and is widely adopted in many continual learning methods [2, 10, 55]. For problem (2), one can utilize any existing memory retrieval methods [2, 3, 10, 55, 65], thus making RAR complimentary and easily integratable with many existing CL methods.

## 3.1 Retrospective Adversarial Perturbation

Existing methods addressing the memory-retrieval problem optimize the model parameters using $\ell_{ER}$ in Eq. 2 where $(X'_M, Y'_M)$ are selected from the buffer $\mathcal{M}$ based on their proposed selection strategies. For the ease of notation, we use $y(x)$ to denote the ground-truth label of sample $x$.

$$\ell_{ER}(X'_M, X_i, \theta) := \ell(X_i, y(X_i); \theta) + \ell(X'_M, y(X'_M); \theta) \tag{2}$$

ER [10] uses naive random sampling to select $X'_M$, GSS [3] encourages gradient diversity while sampling $X'_M$, ASER [55] uses kNN Shapley value to select samples which are representative of $\mathcal{M}$ and close to the samples in $X_i$ in the latent space, and MIR [2] selects replay samples which are maximally interfered/forgotten after the look-ahead probing step. In the look-ahead probing step, the model is updated only on the incoming task samples such that $\theta' = \theta - \eta\nabla_\theta\ell(X_i, y(X_i); \theta)$.

Although the methods mentioned above cover crucial aspects (diversity, representativeness, similarity to incoming batch) for selecting $X'_M$, they do not capture the local pairwise interactions between the memory and current-task samples. Motivated by this, Retrospective Adversarial Replay Loss ($\ell_{RAR}$) finds a cardinality-constrained pairing ($E$) between memory and incoming task samples such that the loss on memory samples that are perturbed towards the paired current-task samples is maximized:

$$\ell_{RAR}(X_M \times X_i, \theta, \theta') := \max_{\substack{E \subset X_M \times X_i \\ |E|=c}} \sum_{(x_M, x_i) \in E} \ell(g_{\theta'_l}(x_M, x_i), y(x_M); \theta) \tag{3}$$

Here, $X_M$ denotes the entire input samples present in $\mathcal{M}$, and $g_{\theta'_l}(x_M, x_i)$ described mathematically in Eq. 4 can be seen as a targeted perturbation of $x_M$ that is visually similar to $x_M$ but close to the latent embedding of target $x_i \in X_i$ in the $l$-th layer latent space modeled by $\theta'$. Since the min-objective in Eq. 4 aims to move the perturbed replay sample close to the current-task sample, $y(x_M) \neq y(x_i)$ for a particular pairing to be considered and to filter out any trivial pairings while optimizing Eq. 3.

$$g_{\theta'_l}(x_M, x_i) = \operatorname*{argmin}_{z} d_{\theta'_l}(z, x_i) \text{ s.t. } ||z - x_M||_\infty \leq \epsilon \tag{4}$$

$$\text{where } d_{\theta'_l}(z, x_i) := ||\theta'_l(z) - \theta'_l(x_i)||_2 \tag{5}$$

Since $x_M$ and $x_i$ have different labels, such perturbed samples that we get from Eq. 4 are confusing to the continual learner in terms of distinguishability from the current-task sample $x_i$, while still representing previous tasks based on their visual similarity to $x_M \in \mathcal{M}$. Eq. 3 essentially helps in the synthesis of targeted adversarial samples that are close to incoming samples $X_i$ in the latent space and hence the learner will miss making the correct prediction $y(X_M)$ despite the fact that the adversarial samples are anchored around the original memory samples $X_M$. Training on such samples can help the model to learn about the boundaries between previous and current tasks and also retain the knowledge of previous tasks [7, 11, 23]. Note that the perturbations are performed based on the look-ahead parameters $\theta'$. In other words, assuming we only learn from the current-task samples $X_i$ and update the parameters to $\theta'$, the generated adversarial samples give us many training samples that show conflicting behaviors on $\theta'$ and get easily forgotten by the model. In retrospect, we can potentially prevent such forgetting when we are still at $\theta$ by using those targeted adversarial samples as training signals, i.e., we minimize the RAR loss (Eq. 3) to update the model $\theta$.

Optimizing Eq. 3 over all feasible pairs of $X_M$ and $X_i$ is computationally challenging as we effectively need to compute the adversarial perturbations on all such pairs to select the most confusing subset incurring the maximum RAR loss. Moreover, each adversarial sample synthesis (Eq. 4) requires multiple gradient steps as we need to run adversarial attack algorithms such as the iterative-FGSM [15, 20] covered in detail in Appendix H. Thus, to eliminate this bottleneck, we propose two pruning strategies covered in sections 3.2 and 3.3 to reduce the candidate anchor-target pairs in $X_M \times X_i$ for adversarial perturbations and approximately optimize Eq. 3.

## 3.2 Selecting Replay Data from Buffer

To reduce the computational overhead, we only consider a subset of the memory buffer $X'_M \subset \mathcal{M}$ instead of considering all memory input samples $X_M$. Memory-retrieval methods such as ER [10], GSS [3], ASER [55], and MIR [2] can be used to select a smaller replay set $X'_M$. This again reiterates that RAR is generic and complementary to existing memory-retrieval methods.

### 3.3 Pairing between Buffer and Current-task Data

To further bring down the computational costs of adversarial perturbations, we perform another pruning over the possible pairs between $X'_M \times X_i$ based on samples' hidden representations. I.e., we first compute the distance $d_{\theta'_l}(x'_M, x_i)$, which is essentially the target value for the adversarial perturbation (Eq. 4) for every pair $(x'_M, x_i) \in X'_M \times X_i$ with different labels $y(x'_M) \neq y(x_i)$ and then screen out the pairs with larger distance values, maintaining only the smaller-distance ones, in accordance with the min-objective of Eq. 4. To utilize the information from the selected memory samples, we pair each $x'_M \in X'_M$ with its nearest neighbor with a different class/label in the incoming batch $X_i$ of current task. Nearest-neighbor-based matched pairs already represent the most confusing samples as the samples in a pair belong to different classes. Thus, it is easier for the optimization procedure approximating Eq. 4 to further reduce the distances between pairs coming from different classes (shown in Fig. 2b), aiding in the synthesis of even more confusing adversarial samples.

**Computational overhead for optimizing Eq. 3**: First, we select memory samples for replay $X'_M$ (Sec. 3.2) which has the same costs as the backbone memory-retrieval method. Next, we compute the distances between latent features of $X'_M$ and incoming batch $X_i$ with computational complexity of $\mathcal{O}(|X'_M| * |X_i|)$. This is minimal given the online nature of CL where the training batch size is often set to 10. After performing the nearest neighbor matching, we optimize Eq. 4 via iterative-FGSM [15] which introduces some additional computational costs, but we keep it marginal by using only two steps of iterative-FGSM. We provide further details in Appendix E and H.

Combining our RAR loss with the loss on selected memory samples and incoming batch samples, we have our overall objective (similar to Eq. 2) defined as follows, where $\beta$ is a hyperparameter to control the trade-offs between the losses on original and adversarially synthesized memory samples.

$$\ell_{all}(X'_M, X_i, \theta) := \ell(X_i, y(X_i); \theta) + \beta \ell_{RAR}(X'_M \times X_i, \theta, \theta') + (1-\beta)\ell(X'_M, y(X'_M); \theta) \quad (6)$$

---

**Algorithm 1 RAR** - Retrospective Adversarial Replay for Continual Learning

---

1: **Input:** Tasks $(\mathcal{D}_1, \mathcal{D}_2, \dots \mathcal{D}_T)$, Model $\theta$, Learning Rate $\eta$, Memory $\mathcal{M}$, Memory Size $m$, Replay Budget $k$, Trade-off Parameter $\beta$, Perturbation Strength $\epsilon$ & number of gradient steps $n$
2: **Output:** Model $\theta$
3: **for** task $i \in [T]$ **do**
4:     **for** $(X_i, Y_i) \sim \mathcal{D}_i$ **do**
5:         $\theta' \leftarrow \text{SGD}(\ell(X_i, Y_i; \theta), \theta, \eta)$ { // Look-ahead update of model parameters}
6:         Select $(X'_M, Y'_M)$ from $\mathcal{M}$ s.t. $|X'_M| = k$ { // using existing memory-retrieval methods}
7:         Let $S := \emptyset$
8:         **for** $x'_M \in X'_M$ **do**
9:             $x'_i \in \text{argmin}_{x_i \in X_i, y(x'_M) \neq y(x_i)} d_{\theta'_l}(x'_M, x_i)$
10:           $S \leftarrow S \cup \{(x'_M, x'_i)\}$
11:         **end for**
12:         $\ell_{RAR}(S, \theta, \theta') \leftarrow \sum_{(x'_M, x_i) \in S} \ell(g_{\theta'_l}(x'_M, x_i), y(x'_M); \theta)$ { // perform the targeted adversarial perturbations (Appendix H) and define the corresponding RAR loss}
13:         $\ell_{all} \leftarrow \ell(X_i, Y_i; \theta) + \beta \, \ell_{RAR}(S, \theta, \theta') + (1-\beta) \, \ell(X'_M, Y'_M; \theta)$
14:         $\theta \leftarrow \text{SGD}(\ell_{all}, \theta, \eta)$
15:         $\mathcal{M} \leftarrow \text{ReservoirUpdate}(X_i, Y_i, \mathcal{M}, m)$
16:     **end for**
17: **end for**

---

**Description of RAR framework**: In Alg. 1, we describe our retrospective adversarial replay algorithm for continual learning. The algorithm essentially consists of three steps: (1) in Lines 5-6, we first perform the look-ahead parameter probing using the incoming samples of the current task to get $\theta'$, and select the subset $X'_M$ using existing memory retrieval methods, such as ER, MIR, ASER, etc. (2) in Lines 7-11, we prune down the pairings between $X'_M$ and $X_i$ based on distances between hidden representations computed using Eq. 5, and (3) in Line 12, we define the RAR loss by performing adversarial perturbations on the selected pairs $S$. Note that since we use the selected pairings $S$ of size $k$ as an approximation of the true maximum-valued pairs, we can just sum over

entries in $S$ instead of picking the maximum subset as done in Eq. 3. In Line 15, we update the memory buffer $\mathcal{M}$ based on incoming samples $(X_i, Y_i)$ using reservoir sampling [62] and keep the memory limit $m$.

## 3.4 MixUp based Retrospective Adversarial Augmentation

We also propose another method *mix-RAR* utilizing the *mixup* technique [67] to bring in more diversity to the pool of replay examples selected from the buffer. After creating virtual examples $(\tilde{x}_M, \tilde{y}_M)$ using $x_i, x_j \in X'_M$ and $\lambda \in [0, 1]$ such that $\tilde{x}_M = \lambda x_i + (1 - \lambda) x_j$ and $\tilde{y}_M = \lambda y_i + (1 - \lambda) y_j$, we follow the original RAR algorithm to achieve pairing and subsequent targeted adversarial perturbation as shown in Alg 5 in Appendix A. The proposed MixUp strategy is different from [21, 40] as we apply mixup among the replay samples and then generate RAR perturbed samples anchored around them. This is consistent with RAR's objective. Given a current-task's sample from class (c), training on an RAR perturbation applied to a mixup between two buffered samples from different past classes (a,b) can potentially help in capturing the forgetting frontier between three pairs of classes (a,c), (b,c) and (a,b), thus minimizing forgetting on both past classes.

# 4 Experiments

**Datasets**: We evaluate RAR on four supervised image classification benchmarks for task-free CL. **(1) Split-MNIST**[31] has five disjoint tasks, each having two classes. **(2) Split-CIFAR10** [28] consists 5 disjoint tasks with each task having two classes and 10k training examples. **(3) Split-CIFAR100** deals with 20 disjoint tasks with each task having 5 classes. **(4) Split-miniImageNet** [14] consists 20 disjoint tasks with each task having 5 classes and 2.5k training examples. We cover further statistics in Appendix D. All these datasets deal with the class-incremental scenario [61]. Our implementation code is available at https://github.com/lillykumari8/RAR-CL.

Table 1: Average Accuracy ($\uparrow$) on different task-free CL datasets across different buffer sizes (m). We average results over 15 runs for all except for Split-MNIST where we average over 20 runs.

| Method | Split-MNIST | | Split-CIFAR10 | | | Split-CIFAR100 | Split-miniImageNet |
|---|---|---|---|---|---|---|---|
| | m=500 | m=1000 | m=200 | m=500 | m=1000 | m=10,000 | m=10,000 |
| Fine-tuning | $19.0 \pm 0.2$ | $19.0 \pm 0.2$ | $18.4 \pm 0.3$ | $18.4 \pm 0.3$ | $18.4 \pm 0.3$ | $3.06 \pm 0.2$ | $2.84 \pm 0.4$ |
| AGEM | $29.02 \pm 5.3$ | - | $22.7 \pm 1.8$ | $22.7 \pm 1.9$ | $22.6 \pm 0.7$ | $2.40 \pm 0.2$ | $2.92 \pm 0.3$ |
| ER | $80.7 \pm 2.1$ | $83.3 \pm 1.4$ | $25.7 \pm 1.3$ | $31.9 \pm 0.9$ | $39.5 \pm 1.7$ | $26.0 \pm 0.3$ | $23.7 \pm 0.5$ |
| ER-RAR (ours) | $\mathbf{84.2 \pm 1.8}$ | $\mathbf{86.9 \pm 1.1}$ | $\mathbf{33.2 \pm 1.4}$ | $\mathbf{40.4 \pm 1.7}$ | $\mathbf{44.0 \pm 1.4}$ | $\mathbf{29.8 \pm 0.4}$ | $\mathbf{27.9 \pm 0.5}$ |
| ER-mix | $80.9 \pm 1.3$ | $81.7 \pm 1.1$ | $24.9 \pm 1.8$ | $33.0 \pm 2.3$ | $37.5 \pm 1.6$ | $26.6 \pm 0.8$ | $23.0 \pm 1.1$ |
| ER-mix-RAR (ours) | $\mathbf{86.1 \pm 1.1}$ | $\mathbf{86.9 \pm 1.0}$ | $\mathbf{37.2 \pm 1.0}$ | $\mathbf{41.6 \pm 1.0}$ | $\mathbf{44.4 \pm 0.8}$ | $\mathbf{31.5 \pm 0.6}$ | $\mathbf{28.8 \pm 0.5}$ |
| MIR | $84.8 \pm 1.2$ | $86.3 \pm 1.4$ | $28.0 \pm 1.5$ | $37.5 \pm 1.6$ | $45.3 \pm 1.2$ | $26.7 \pm 0.4$ | $24.3 \pm 0.6$ |
| MIR-RAR (ours) | $\mathbf{87.9 \pm 1.4}$ | $\mathbf{89.4 \pm 1.1}$ | $\mathbf{33.3 \pm 1.1}$ | $\mathbf{43.1 \pm 1.1}$ | $\mathbf{45.3 \pm 1.5}$ | $\mathbf{29.3 \pm 0.4}$ | $\mathbf{26.7 \pm 0.7}$ |
| MIR-mix | $85.7 \pm 0.8$ | $86.6 \pm 0.7$ | $35.2 \pm 1.2$ | $40.7 \pm 0.8$ | $42.5 \pm 1.5$ | $27.6 \pm 0.5$ | $23.3 \pm 0.8$ |
| MIR-mix-RAR (ours) | $\mathbf{89.0 \pm 0.7}$ | $\mathbf{89.1 \pm 1.0}$ | $\mathbf{37.6 \pm 1.0}$ | $\mathbf{42.9 \pm 1.1}$ | $\mathbf{44.7 \pm 1.3}$ | $\mathbf{29.9 \pm 0.4}$ | $\mathbf{27.7 \pm 0.5}$ |
| ASER | $79.6 \pm 2.5$ | $79.5 \pm 1.5$ | $24.2 \pm 1.2$ | $32.2 \pm 1.5$ | $37.0 \pm 2.3$ | $26.5 \pm 0.6$ | $25.0 \pm 0.8$ |
| ASER-RAR (ours) | $\mathbf{81.3 \pm 1.5}$ | $\mathbf{83.2 \pm 1.1}$ | $\mathbf{34.5 \pm 1.2}$ | $\mathbf{40.4 \pm 1.0}$ | $\mathbf{42.7 \pm 1.3}$ | $\mathbf{29.5 \pm 0.4}$ | $\mathbf{26.9 \pm 0.7}$ |
| ASER-mix | $80.2 \pm 1.8$ | $79.6 \pm 1.7$ | $25.9 \pm 1.2$ | $29.5 \pm 1.8$ | $34.0 \pm 1.7$ | $26.8 \pm 0.6$ | $22.1 \pm 0.9$ |
| ASER-mix-RAR (ours) | $\mathbf{82.0 \pm 1.6}$ | $\mathbf{84.0 \pm 1.2}$ | $\mathbf{35.6 \pm 0.9}$ | $\mathbf{39.2 \pm 1.4}$ | $\mathbf{41.5 \pm 1.6}$ | $\mathbf{29.9 \pm 0.3}$ | $\mathbf{27.5 \pm 0.5}$ |
| iid-online | $86.8 \pm 1.1$ | $86.8 \pm 1.1$ | $60.8 \pm 1.0$ | $60.8 \pm 1.0$ | $60.8 \pm 1.0$ | $18.13 \pm 0.8$ | $17.53 \pm 1.6$ |
| iid-offline | $92.3 \pm 0.5$ | $92.3 \pm 0.5$ | $79.2 \pm 0.4$ | $79.2 \pm 0.4$ | $79.2 \pm 0.4$ | $42.0 \pm 0.9$ | $37.46 \pm 1.3$ |

**Evaluation Metrics**: Following [8, 37], we use two standard metrics used in CL to evaluate RAR's performance. *Final average accuracy ($A_T$)* assesses the overall performance of the model whereas *Final average forgetting ($F_T$)* measures how much the model has forgotten each task once the online learning has completed. Lower forgetting is better. Below, $a_{i,j}$ denotes the accuracy evaluated on the test set of task $j$ after the model has experienced all tasks up to $i$.

$$A_T = \frac{1}{T} \sum_{j=1}^{T} a_{T,j} \quad F_T = \frac{1}{T-1} \sum_{j=1}^{T-1} \max_{l \in \{1, \dots T-1\}} (a_{l,j} - a_{T,j}) \tag{7}$$

**Settings and Hyperparameters**: We follow the same setup as [2] for deciding model architectures for all four datasets. For Split-MNIST, we use an MLP classifier with two hidden layers, each with

Table 2: Average Forgetting ($\downarrow$) on different task-free CL datasets across different buffer sizes (m). We average results over 15 runs for all except for Split-MNIST where we average over 20 runs.

| | Split-MNIST | | Split-CIFAR10 | | | Split-CIFAR100 | Split-miniImageNet |
|---|---|---|---|---|---|---|---|
| **Method** | m=500 | m=1000 | m=200 | m=500 | m=1000 | m=10,000 | m=10,000 |
| ER | $17.1 \pm 2.6$ | $13.1 \pm 1.8$ | $51.3 \pm 4.1$ | $39.0 \pm 2.1$ | $26.3 \pm 2.4$ | $17.1 \pm 0.5$ | $28.3 \pm 0.6$ |
| ER-RAR (ours) | $\mathbf{11.0 \pm 2.5}$ | $\mathbf{7.4 \pm 1.1}$ | $\mathbf{37.6 \pm 2.1}$ | $\mathbf{25.1 \pm 2.5}$ | $\mathbf{18.1 \pm 2.0}$ | $\mathbf{10.8 \pm 0.4}$ | $\mathbf{16.4 \pm 0.8}$ |
| ER-mix | $17.3 \pm 1.6$ | $15.8 \pm 1.5$ | $54.9 \pm 5.2$ | $37.2 \pm 3.5$ | $25.8 \pm 4.0$ | $21.9 \pm 1.1$ | $33.9 \pm 1.5$ |
| ER-mix-RAR (ours) | $\mathbf{9.6 \pm 1.4}$ | $\mathbf{8.4 \pm 1.5}$ | $\mathbf{27.4 \pm 2.3}$ | $\mathbf{20.1 \pm 1.6}$ | $\mathbf{15.4 \pm 1.1}$ | $\mathbf{10.7 \pm 0.4}$ | $\mathbf{16.5 \pm 1.1}$ |
| MIR | $10.7 \pm 1.6$ | $7.5 \pm 1.8$ | $48.1 \pm 2.9$ | $32.2 \pm 2.1$ | $19.1 \pm 1.8$ | $13.8 \pm 0.4$ | $21.4 \pm 0.6$ |
| MIR-RAR (ours) | $\mathbf{6.1 \pm 1.2}$ | $\mathbf{3.1 \pm 0.7}$ | $\mathbf{33.5 \pm 2.2}$ | $\mathbf{21.1 \pm 2.2}$ | $\mathbf{14.1 \pm 1.6}$ | $\mathbf{10.9 \pm 0.5}$ | $\mathbf{15.5 \pm 0.8}$ |
| MIR-mix | $8.7 \pm 0.9$ | $7.8 \pm 1.1$ | $\mathbf{24.7 \pm 2.6}$ | $\mathbf{16.1 \pm 2.2}$ | $\mathbf{12.9 \pm 2.7}$ | $16.4 \pm 0.6$ | $30.3 \pm 1.2$ |
| MIR-mix-RAR (ours) | $\mathbf{4.6 \pm 1.0}$ | $\mathbf{4.1 \pm 1.0}$ | $26.6 \pm 1.9$ | $17.4 \pm 1.7$ | $13.1 \pm 2.0$ | $\mathbf{11.3 \pm 0.5}$ | $\mathbf{13.7 \pm 1.0}$ |
| ASER | $19.1 \pm 3.0$ | $17.8 \pm 2.4$ | $64.3 \pm 3.3$ | $45.5 \pm 3.3$ | $29.3 \pm 3.4$ | $16.8 \pm 0.6$ | $27.0 \pm 1.0$ |
| ASER-RAR (ours) | $\mathbf{15.4 \pm 2.2}$ | $\mathbf{12.0 \pm 1.9}$ | $\mathbf{35.4 \pm 2.0}$ | $\mathbf{21.4 \pm 2.0}$ | $\mathbf{14.5 \pm 1.6}$ | $\mathbf{8.9 \pm 0.4}$ | $\mathbf{15.9 \pm 1.0}$ |
| ASER-mix | $18.6 \pm 2.2$ | $18.9 \pm 2.6$ | $54.7 \pm 3.8$ | $50.1 \pm 4.7$ | $38.1 \pm 3.3$ | $21.7 \pm 0.8$ | $33.7 \pm 1.0$ |
| ASER-mix-RAR (ours) | $\mathbf{16.1 \pm 2.3}$ | $\mathbf{12.0 \pm 1.7}$ | $\mathbf{30.3 \pm 1.8}$ | $\mathbf{21.5 \pm 3.3}$ | $\mathbf{18.7 \pm 3.2}$ | $\mathbf{11.0 \pm 0.5}$ | $\mathbf{12.4 \pm 1.0}$ |

400 units with ReLU activation, followed by a linear classifier layer with 10 units. For Split CIFAR-10, Split CIFAR-100, and Split mini-ImageNet, we use a reduced ResNet-18 classifier [22]. The replay budget $k$ is the same as the mini-batch size (fixed to 10) irrespective of the buffer size $m$. For hyperparameters tuning on each dataset, we hold-out $5\%$ of the training samples for each task and use it as a validation set. We provide additional details about the implementation settings in Appendix F.

## 4.1 Main Results

We report the average accuracy and forgetting in Table 1 and 2 respectively for experiments on all four datasets under various buffer size constraints. We use three different memory-retrieval methods, namely *ER* [10], *MIR* [2], and *ASER* [55] and compare *RAR* and *mix-RAR* with their respective baselines which do not use the RAR loss (Eq. 3). For fair comparison, we compare *mix-RAR* with baseline retrieval methods using mixup between replay samples. We also compare against standard baseline methods: (a) *Fine-tuning*, which trains the model continuously on the stream of tasks without employing any strategy to avoid forgetting, (b) *Averaged Gradient*

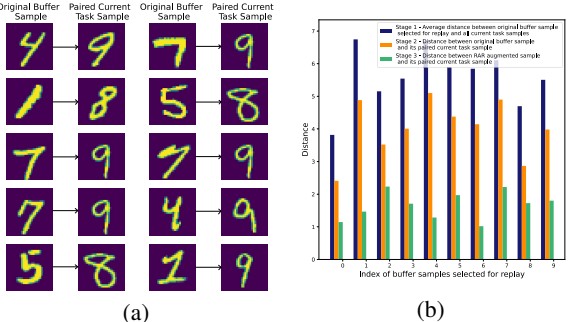

(a)                    (b)

Figure 2: Split-MNIST - (a) Buffer samples selected for replay using MIR strategy along with their paired current-task sample. (b) Visualization of how the distance between the buffered replay samples and its nearest neighbor in the current tasks' batch evolves across different stages of RAR

*Episodic Memory (AGEM)* [9] (c) *iid-online*, which trains a model on the iid-data sampled from entire dataset $\mathcal{D} = \cup_{i=1}^{T} \mathcal{D}_i$ for a single epoch, (d) *iid-offline*, which trains a model similar to the iid-online setting but for multiple epochs (three epochs considered in the reported results). We compare against another memory sample perturbation method called *GMED* [25] in Table 3 to avoid repetition.

We provide the pseudo-code for ER (Alg. 2), ER-RAR (Alg. 3), MIR-RAR (Alg. 4), and mix-RAR (Alg. 5) in Appendix A to describe how our proposed framework can be unified with existing CL methods. Also, we leave out the results against GSS [3] as both MIR [2] and ASER [55] outperform it in terms of accuracy and forgetting. As shown in Table 1, both *RAR* and *mix-RAR* using different memory-retrieval methods consistently outperform their respective non-RAR baselines (without and with mixup) across all four datasets. From Table 2, we observe that the improvements obtained by integrating *RAR* with respective replay selection methods *{ER, MIR & ASER}* (both without and with mixup) are significant. In case of Split-CIFAR10, *MIR-mix* tends to perform slightly better than *MIR-mix-RAR* in terms of forgetting, but based on accuracy improvement that *MIR-mix-RAR* brings, the relative difference is negligible since forgetting is dependent on the maximum accuracy that a continual learner achieves across different tasks.

In the case of Split-miniImageNet, ER-mix-RAR method achieves the best average accuracy of $28.8\%$, outperforming other MIR & ASER based methods, and has similar observation on the Split-CIFAR100 dataset. We hypothesize that for large buffer sizes such as 10k (100 examples per

class), the ER-based selection aids in sampling a diverse set of classes from the memory buffer compared to MIR & ASER, thus improving the overall accuracy. However, since MIR & ASER consider samples suffering interference from the new task's data, MIR-mix-RAR & ASER-mix-RAR help minimize the forgetting on the old tasks.

The improved performance of ER-RAR, MIR-RAR, and ASER-RAR compared to ER, RAR, and ASER respectively demonstrates that RAR is able to alleviate catastrophic forgetting on old tasks via generating diverse and confusing samples close to the forgetting frontier. To ensure fair comparison in terms of computational costs, in Table 5 in Appendix I, we also compare RAR to GMED [25], ER-3-iters, and MIR-3-iters. 3-iters means that the model is trained for multiple iterations on the incoming batch and buffer samples as a new batch comes in.

Table 3: Comparison of RAR to other replay data perturbation methods: (a) Random edit using same perturbation tolerance as RAR (b) GMED. Numbers within bracket denote overall buffer size used.

| Method | Split-MNIST (200) | | Split-CIFAR10 (200) | | Split-CIFAR10 (500) | | Split-CIFAR100 (10k) | |
|---|---|---|---|---|---|---|---|---|
| | Accuracy | Forgetting | Accuracy | Forgetting | Accuracy | Forgetting | Accuracy | Forgetting |
| ER-Random | $80.2 \pm 2.3$ | $18.7 \pm 3.1$ | $28.5 \pm 1.6$ | $41.9 \pm 2.5$ | $38.3 \pm 1.1$ | $30.2 \pm 1.6$ | $26.3 \pm 0.5$ | $14.4 \pm 0.5$ |
| ER-GMED | $82.5 \pm 1.4$ | $15.0 \pm 1.7$ | $30.4 \pm 0.9$ | $39.0 \pm 1.3$ | $38.3 \pm 1.1$ | $27.7 \pm 1.3$ | $25.5 \pm 0.5$ | $13.5 \pm 0.5$ |
| ER-RAR (ours) | $\mathbf{84.0 \pm 1.1}$ | $\mathbf{11.8 \pm 1.5}$ | $\mathbf{33.2 \pm 1.4}$ | $\mathbf{37.6 \pm 2.1}$ | $\mathbf{40.4 \pm 1.7}$ | $\mathbf{25.1 \pm 2.5}$ | $\mathbf{29.8 \pm 0.4}$ | $\mathbf{10.8 \pm 0.4}$ |
| MIR-Random | $82.3 \pm 1.6$ | $15.9 \pm 1.7$ | $29.1 \pm 1.2$ | $45.3 \pm 2.4$ | $39.2 \pm 1.3$ | $29.0 \pm 2.1$ | $26.5 \pm 0.3$ | $13.6 \pm 0.5$ |
| MIR-GMED | $84.2 \pm 1.2$ | $12.2 \pm 1.0$ | $27.5 \pm 1.3$ | $45.5 \pm 1.6$ | $37.2 \pm 1.1$ | $28.6 \pm 1.9$ | $25.1 \pm 0.3$ | $11.7 \pm 0.4$ |
| MIR-RAR (ours) | $\mathbf{86.2 \pm 0.6}$ | $\mathbf{9.1 \pm 0.8}$ | $\mathbf{33.3 \pm 1.1}$ | $\mathbf{33.5 \pm 2.2}$ | $\mathbf{43.1 \pm 1.1}$ | $\mathbf{21.1 \pm 2.2}$ | $\mathbf{29.3 \pm 0.4}$ | $\mathbf{10.9 \pm 0.5}$ |

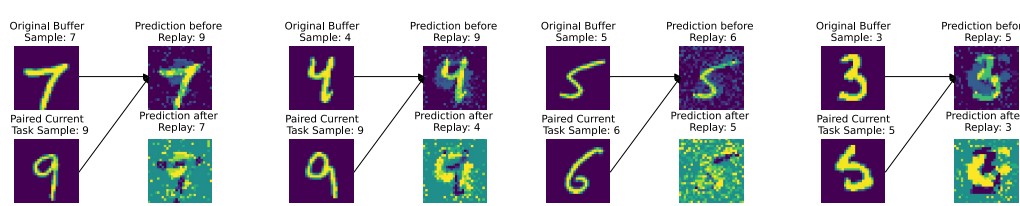

Figure 3: Retrospective Adversarial Perturbations (top-right of each sub-figure) for different replay samples shown along side original replay samples (top-left) and paired current-task sample (bottom-left). The bottom-right image shows the scaled perturbation that was added to the original buffered sample to generate the RAR perturbed data.

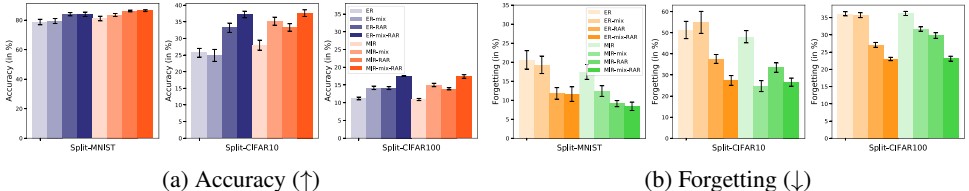

(a) Accuracy (↑)              (b) Forgetting (↓)

Figure 4: Results under small buffer constraints, 20 examples per class for each dataset

**Case study: Visualizing the Three Stages of RAR** For the Split-MNIST dataset, we visually analyze the three stages of the RAR framework for a buffer size of 500 using MIR as the memory-retrieval method. Fig. 2a shows the samples selected for replay using the MIR score (Eq. 8) and their paired samples from the current task's batch based on distances computed using Eq. 5. Fig. 2b shows how the distance between a replaying sample and current task's data changes over the three stages: (1) for each sample selected from the buffer, the blue bar represents its average distance to all samples in the current-task batch; (2) after pairing each buffered replay sample with its nearest neighbor as described in Section 3.3, its distance to the paired sample (orange bar) becomes smaller than the blue bar; (3) after applying adversarial perturbations to each buffered sample as described in Section 3.1, the distance is further reduced. The decrease of distance indicates that the pairing and adversarial perturbation in RAR are effective in generating samples closer to the forgetting frontier than the original buffer data selected for replay, meaning the RAR perturbations are similar to $X_M$ but very close to the paired samples in $X_i$ (current task's batch) in the *l-th* layer latent representation space of model $\theta$. In Fig. 3, we show four examples of RAR perturbed samples generated via Eq. 4 from four buffer samples and their paired samples from the current task. For most of the buffer samples, a single gradient step of $\theta$

replaying them and their RAR perturbed counterparts suffices to correct their wrong predictions (e.g., the current task's classes), hence effectively alleviating the model's forgetting on previous tasks.

## 4.2 Advantage of MixUp in Small-Buffer Cases

On each dataset, as reported in Table 2, all mix-RAR methods perform better compared to RAR methods in terms of reducing forgetting on previous tasks with slight improvement in average accuracy. The proposed use of MixUp among the replay samples before RAR is able to increase the variation of RAR perturbed samples, which is essential to CL with a small buffer. To verify its effectiveness, we conduct experiments that store only 20 examples per class. Figure 4 shows that ER-mix-RAR & MIR-mix-RAR consistently outperform other ER & MIR methods respectively. We can achieve even more significant improvements in forgetting, particularly for more challenging tasks such as Split-CIFAR100.

## 4.3 Ablation Study to Verify the Effectiveness of RAR

In Table 3, we compare RAR methods to two baseline methods: (a) *Random* which add a perturbation of the same tolerance as the RAR method to the replay samples before the experience replay stage, (b) *GMED* [25] which performs smart gradient updates in the input space by treating each replay sample individually (no local pairing considered between buffer and current task's data). As can be seen in Table 3, ER-RAR and MIR-RAR consistently outperform other methods confirming RAR's superior ability to tackle catastrophic forgetting via generating targeted perturbations influenced by the local interference between different task's data.

## 4.4 Sensitivity Analysis of Hyperparameters

In Fig. 5, we plot the sensitivity of RAR's performance with respect to the number of gradient steps $n$ used to generate the retrospective adversarial samples via optimizing Eq. 4 in the input space. For Split-CIFAR10 and Split-miniImageNet (where we report results using two gradient steps), we observe that for a larger number of steps {5, 10}, the performance stays comparable. This suggests that nearest-neighbor-based pairing using distance computations (covered in Sec 3.3) helps achieve anchor-target pairs which are already optimized and hence do not require numerous gradient updates to get classified as the paired target class from the current task.

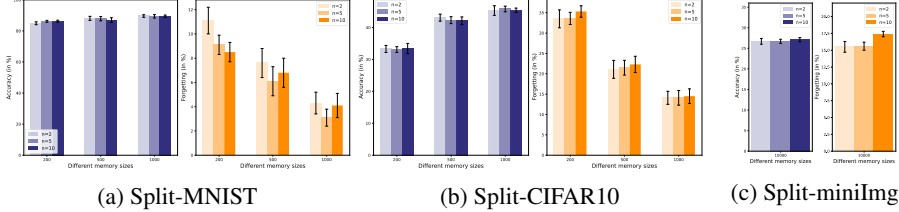

(a) Split-MNIST        (b) Split-CIFAR10        (c) Split-miniImg

Figure 5: Sensitivity of the performance of RAR (using MIR) to number of perturbation steps $(n)$ used to generate the RAR perturbed samples on different datasets under various buffer constraints.

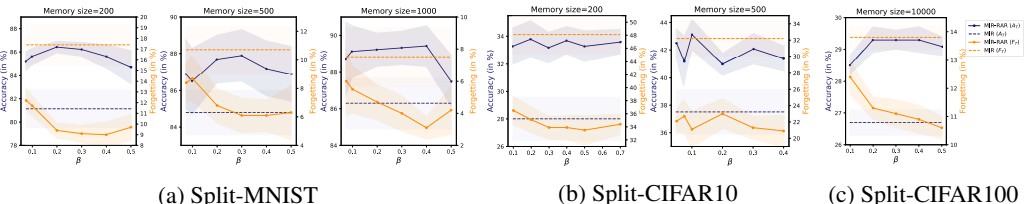

(a) Split-MNIST        (b) Split-CIFAR10        (c) Split-CIFAR100

Figure 6: Sensitivity of the performance of RAR (using MIR) to the trade-off coefficient $(\beta)$ dedicated to $\ell_{RAR}$ on different datasets under various buffer constraints.

Next, we study the sensitivity of RAR corresponding to $\beta$ which denotes the trade-off coefficient associated with $\ell_{RAR}$ in the final loss objective Eq. 6 — we use only one coefficient to reduce hyperparameter search costs. Fig. 6 shows that RAR (using MIR) outperforms vanilla MIR in terms of both accuracy (solid blue line is always above the dashed blue line) and forgetting (solid orange line always stays below the dashed orange line) for various non-zero values of $\beta$ across the studied datasets.

# 5 Conclusion

In this paper, we study how to perturb limited buffer replay data to mitigate catastrophic forgetting in continual learning. We propose "Retrospective Adversarial Replay (RAR)", which focuses on previous tasks' data near the forgetting boundary to the current task in the model's representation space. RAR first allocates the most confusing pairs of a buffered sample and a current task's sample and then applies a bounded targeted adversarial perturbation to the former, which further moves it towards the latter in the model's representation space and creates a perturbed sample close to the boundary. Moreover, we study the role of MixUp in increasing the variation of replay augmentations, which significantly improves CL in the small buffer regime. When integrated with existing replay methods, RAR consistently outperforms previous CL methods across benchmark datasets under different settings.

## Acknowledgements

This work was supported in part by the CONIX Research Center, one of six centers in JUMP, a Semiconductor Research Corporation (SRC) program sponsored by DARPA. We would also like to thank the Melodi lab students: Arnav Das and Gantavya Bhatt for their useful discussions.

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
