# A Details of RAR based algorithms

**Standard ER**    In Algorithm 2, we present the standard experience replay method [10, 49].

---

**Algorithm 2 ER** - Experience Replay based Continual Learning

---
1: **Input:** Tasks $(\mathcal{D}_1, \mathcal{D}_2, \ldots \mathcal{D}_T)$, Model $\theta$, Learning Rate $\eta$, Memory $\mathcal{M}$, Memory Size $m$, Replay Budget $k$
2: **Output:** Model $\theta$
3: **for** task $i \in [T]$ **do**
4:   **for** $(X_i, Y_i) \sim \mathcal{D}_i$ **do**
5:     $(X'_M, Y'_M) \sim \mathcal{M}$ { // randomly select $k$ samples from memory}
6:     $\ell_{ER} \leftarrow \ell(X_i, Y_i; \theta) + \ell(X'_M, Y'_M; \theta)$
7:     $\theta \leftarrow \text{SGD}(\ell_{ER}, \theta, \eta)$
8:     $\mathcal{M} \leftarrow \text{ReservoirUpdate}(X_i, Y_i, \mathcal{M}, m)$
9:   **end for**
10: **end for**

---

**ER-RAR**    In the ER-RAR method, we simply modify the replay sample selection step (line 6) of Alg. 1. As shown in line 6 of Algorithm 3, we rely on random sampling to get a subset of the buffer for replay.

---

**Algorithm 3 ER-RAR** - Retrospective Adversarial Replay for Continual Learning using vanilla ER as memory-retrieval method

---
1: **Input:** Tasks $(\mathcal{D}_1, \mathcal{D}_2, \ldots \mathcal{D}_T)$, Model $\theta$, Learning Rate $\eta$, Memory $\mathcal{M}$, Memory Size $m$, Replay Budget $k$, Trade-off Parameter $\beta$, Perturbation Strength $\epsilon$ & number of gradient steps $n$
2: **Output:** Model $\theta$
3: **for** task $i \in [T]$ **do**
4:   **for** $(X_i, Y_i) \sim \mathcal{D}_i$ **do**
5:     $\theta' \leftarrow \text{SGD}(\ell(X_i, Y_i; \theta), \theta, \eta)$ { // Look-ahead update of model parameters}
6:     $(X'_M, Y'_M) \sim \mathcal{M}$ { // randomly select $k$ samples from memory}
7:     Let $S := \emptyset$
8:     **for** $x'_M \in X'_M$ **do**
9:       $x'_i \in \text{argmin}_{x_i \in X_i, y(x'_M) \neq y(x_i)} d_{\theta'_l}(x'_M, x_i)$.
10:      $S \leftarrow S \cup \{(x'_M, x'_i)\}$.
11:    **end for**
12:    $\ell_{RAR}(S, \theta, \theta') \leftarrow \sum_{(x'_M, x_i) \in S} \ell(g_{\theta'_l}(x'_M, x_i), y(x'_M); \theta)$ { // perform the targeted adversarial perturbations (Appendix H) and define the corresponding loss}
13:    $\ell_{all} \leftarrow \ell(X_i, Y_i; \theta) + \beta\, \ell_{RAR}(S, \theta, \theta') + (1 - \beta)\, \ell(X'_M, Y'_M; \theta)$
14:    $\theta \leftarrow \text{SGD}(\ell_{all}, \theta, \eta)$
15:    $\mathcal{M} \leftarrow \text{ReservoirUpdate}(X_i, Y_i, \mathcal{M}, m)$
16:  **end for**
17: **end for**

---

**Discussion of MIR score and its intuition** (find the buffered data suffering from the most amount of forgetting) The idea of probing to the next-step parameters $\theta'$ that we perform in RAR is similar to the Maximally Interfered Retrieval (MIR) method for memory/buffer retrieval [2], where we compute forgetting scores on the memory samples to identify a subset of memory samples that are negatively impacted by the current-task samples. The forgetting score of MIR for every sample in the memory buffer $x_M$ is defined as follows:

$$\text{MIR}(x_M, \theta, \theta') := \ell(x_M, y(x_M); \theta') - \ell(x_M, y(x_M); \theta) \tag{8}$$

Then a subset of memory samples with the largest MIR scores i.e. having the greatest loss increase from $\theta$ to $\theta'$ are selected. We will denote such a subset as $X'_M$, i.e.

$$X'_M \in \underset{X \subseteq X_M, |X| \leq k}{\text{argmax}} \sum_{x \in X} \text{MIR}(x, \theta, \theta'). \tag{9}$$

**Algorithm 4 MIR-RAR** - Retrospective Adversarial Replay for Continual Learning using MIR as memory-retrieval method

---

1: **Input:** Tasks $(\mathcal{D}_1, \mathcal{D}_2, \ldots \mathcal{D}_T)$, Model $\theta$, Learning Rate $\eta$, Memory $\mathcal{M}$, Memory Size $m$, Replay Budget $k$, Subset Size $c$, Trade-off Parameter $\beta$, Perturbation Strength $\epsilon$ & number of gradient steps $n$
2: **Output:** Model $\theta$
3: **for** task $i \in [T]$ **do**
4:    **for** $(X_i, Y_i) \sim \mathcal{D}_i$ **do**
5:      $\theta' \leftarrow \text{SGD}(\ell(X_i, Y_i; \theta), \theta, \eta)$ { // Look-ahead update of model parameters}
6:      $(X_M, Y_M) \sim \mathcal{M}$ { // randomly select $c$ samples from memory}
7:      Compute $\text{MIR}(x_M, \theta, \theta')$ for $x_M \in X_M$ and select the subset $X'_M$ based on Eq. 9 s.t. $|X'_M| = k$
8:      Let $S := \emptyset$.
9:      **for** $x'_M \in X'_M$ **do**
10:        $x'_i \in \text{argmin}_{x_i \in X_i, y(x'_M) \neq y(x_i)} d_{\theta'_l}(x'_M, x_i)$.
11:        $S \leftarrow S \cup \{(x'_M, x'_i)\}$.
12:      **end for**
13:      $\ell_{RAR}(S, \theta, \theta') \leftarrow \sum_{(x'_M, x_i) \in S} \ell(g_{\theta'_l}(x'_M, x_i), y(x'_M); \theta)$ { // perform the targeted adversarial perturbations (Appendix H) and define the corresponding loss}
14:      $\ell_{all} \leftarrow \ell(X_i, Y_i; \theta) + \beta \, \ell_{RAR}(S, \theta, \theta') + (1 - \beta) \, \ell(X'_M, Y'_M; \theta)$
15:      $\theta \leftarrow \text{SGD}(\ell_{all}, \theta, \eta)$
16:      $\mathcal{M} \leftarrow \text{ReservoirUpdate}(X_i, Y_i, \mathcal{M}, m)$
17:    **end for**
18: **end for**

---

Intuitively, $X'_M$ contains samples that are most confusing with the incoming ones, and thus using such samples as the anchor samples would result in more successful adversarial perturbations.

As we need to compute $\theta'$ for the retrospective adversarial samples, the MIR score does not incur extra computations. Moreover, we can also utilize the subset $X'_M$ as additional training signals similar to the MIR approach, which allows us to trade-off between the adversarially perturbed data points versus real data points that are uniformly sampled from previous tasks.

**Algorithm 5 mix-RAR** - Retrospective Adversarial Replay for Continual Learning using MixUp

---

1: **Input:** Tasks $(\mathcal{D}_1, \mathcal{D}_2, \ldots \mathcal{D}_T)$, Model $\theta$, Learning Rate $\eta$, Memory $\mathcal{M}$, Memory Size $m$, Replay Budget $k$, Subset Size $c$, Trade-off Parameter $\beta$, Perturbation Strength $\epsilon$ & number of gradient steps $n$
2: **Output:** Model $\theta$
3: **for** task $i \in [T]$ **do**
4:    **for** $(X_i, Y_i) \sim \mathcal{D}_i$ **do**
5:      $\theta' \leftarrow \text{SGD}(\ell(X_i, Y_i; \theta), \theta, \eta)$ { // Look-ahead update of model parameters}
6:      Select $(X'_M, Y'_M)$ from $\mathcal{M}$ s.t. $|X'_M| = k$ { // using existing memory-retrieval methods}
7:      $\tilde{X}'_M, \tilde{Y}'_M = \text{mixup}(X'_M, Y'_M, \alpha)$ { // perform mixup on replay samples}
8:      Let $S := \emptyset$
9:      **for** $\tilde{x}_M \in \tilde{X}'_M$ **do**
10:        $x'_i \in \text{argmin}_{x_i \in X_i, y(\tilde{x}_M) \neq y(x_i)} d_{\theta'_l}(\tilde{x}_M, x_i)$.
11:        $S \leftarrow S \cup \{(\tilde{x}_M, x'_i)\}$.
12:      **end for**
13:      $\ell_{RAR}(S, \theta, \theta') \leftarrow \sum_{(\tilde{x}_M, x_i) \in S} \ell(g_{\theta'_l}(\tilde{x}_M, x_i), y(\tilde{x}_M); \theta)$ { // perform the adversarial perturbations (Appendix H) and define the corresponding loss}
14:      $\ell_{all} \leftarrow \ell(X_i, Y_i; \theta) + \beta \, \ell_{RAR}(S, \theta, \theta') + (1 - \beta) \, \ell(X'_M, Y'_M; \theta)$
15:      $\theta \leftarrow \text{SGD}(\ell_{all}, \theta, \eta)$
16:      $\mathcal{M} \leftarrow \text{ReservoirUpdate}(X_i, Y_i, \mathcal{M}, m)$
17:    **end for**
18: **end for**

---

**MIR-RAR** In Algorithm 4, we change line 6 of Algorithm 1 to use the MIR score to select the replay subset $X'_M$.

**mix-RAR** In the mix-RAR method detailed in Algorithm 5, after selecting the replay samples $X'_M$ (using ER, MIR, ASER, etc.) from the memory $\mathcal{M}$, we simply apply MixUp [67] to perform perturbations/editing on the replay examples. Mixup generates virtual training samples $\tilde{x}_M, \tilde{y}_M$ by taking convex combinations of pairs of inputs and their labels.

$$\tilde{x}_M = \lambda x_i + (1 - \lambda)x_j \qquad \tilde{y}_M = \lambda y_i + (1 - \lambda)y_j \qquad (10)$$

where $x_i, x_j \in X'_M$ and $y_i, y_j \in Y'_M$ and $\lambda \in [0, 1]$ is drawn from a Beta distribution $\lambda \sim \text{Beta}(\alpha, \alpha)$, where $\alpha$ is a hyperparameter.

We, then pair up these virtual samples with the current's task samples based on the distance function defined in Eq. 5 and use Eq. 4 to perturb them such that in the representation space modeled by $\theta'_l$, the augmentation of the virtual samples are closer to the paired sample from the current task, than the source/anchor samples $x_i$ and $x_j$.

## B   Details of Related Work

We include detailed explanations about prior works which are relevant in the context of task-free continual learning:

- **Experience Replay (ER)** [10, 49]: ER maintains previously seen examples in a limited size constrained memory buffer for future replay. It deals with the problem of memory retrieval and can be used in both task-free as well as task-dependent continual learning settings, but in all experiments in this paper, we use it as a task-free CL method. As a new incoming batch from current task comes in, we draw a subset of examples from the buffer such that the size of the drawn subset is same as the incoming batch size. In all experiments, we fix this subset size (also referred to as the replay budget) as 10.

- **Gradient Episodic Memory (GEM)** [37]: GEM also maintains an episodic memory for each observed task. As a new incoming batch comes in, GEM projects the gradient of the model parameters such that loss incurred on the episodic memory maintained for each task does not increase while learning a new task. This approach, thus needs access to the task-ids, and is computationally intensive owing to solving a QP program at each iteration of training.

- **Averaged Gradient Episodic Memory (A-GEM)** [9]: Averaged-GEM is an improvement over GEM in terms of relaxing the need to require task-ids. In stead of constraining the model parameters using the entire stored buffer, it randomly samples a subset of examples from the memory buffer to regularize the model parameters in order to minimize forgetting on previously seen tasks.

- **Gradient-based Sample Selection (GSS)** [3]: GSS deals with the problem of Memory Update in task-free CL setting. It aims to maximize samples diversity in the gradient space while selecting examples for memory update. The performance improvement observed is significant when different tasks data are imbalanced. In all experiments covered in this paper, we have used reservoir sampling as the memory update method over all compared SoTA methods to ensure fair comparison.

- **Maximally Interfered Retrieval (MIR)** [2]: MIR is another memory retrieval method which does not need access to task-ids while selecting buffer exemplars for replay. It proposes to first perform a virtual update (one-step look-ahead update) only on the incoming batch of the current batch. It then computes the interference on the memory samples in terms of the loss increase observed w.r.t. the virtual updated model parameters, and selects those buffer samples for which the loss increase is maximum — it selects the maximum interfered/forgotten samples from the buffer. The implementation code provided by the authors that we have used is located at https://github.com/optimass/Maximally_Interfered_Retrieval.

- **Adversarial Continual Learning (ACL)** [16]: ACL focuses on learning two kind of features for task-dependent CL: (1) shared task-invariant (2) task specific features. The architecture grows as a new task comes by adding a task-specific module. It utilizes adversarial learning to train a shared feature encoder and a discriminator using the classic

minimax optimization objective (as used by GANs). The role of the discriminator is to correctly classify the encoded features by their task labels/ids while the role of the encoder is to generate task-invariant features that can fool the discriminator. RAR, on the other hand, focuses on the static architecture task-free CL and utilizes the idea of adversarial perturbations in the input space to perturb the memory replay samples such that they are located close to the forgetting frontier w.r.t. to the current tasks.

- **Adversarial Shapley Value Experience Replay (ASER)** [55]: ASER proposes a scoring method based on kNN Shapley Value (SV) to select memory samples which are representative of the stored memory samples and adversarially close to the decision boundaries of new classes. On the other hand, RAR's objective is to minimize the worst-case experience replay loss by generating targeted adversarial replay perturbations that are visually similar to the chosen replay samples but the CL model mistakes them for the nearest class (from the incoming batch) based on their distance from the decision boundaries. When we apply RAR's objective on ASER chosen replay samples (i.e., compute $X'_M$ in Sec 3.2 using Adversarial SV), both accuracy and forgetting improve quite significantly (Table 1, 2), suggesting that learning on the RAR's adversarial perturbations helps the model to learn about the boundaries between past and current tasks. Also, we use Adversarial SV only for memory retrieval in order to ensure fair comparison across all methods. The implementation code provided by the authors is at https://github.com/RaptorMai/online-continual-learning.

- **Gradient-based Memory Editing for Continual Learning (GMED)** [25]: GMED uses gradient updates to edit/perturb the stored replay samples, to create more challenging samples for rehearsal in a task-free setting. This work is most relevant to RAR. While their objective aims to perturb samples individually to generate increased losses in the upcoming model updates, our method looks into the local pairwise relationships between the most interfered memory samples and the new task's samples. By utilizing such information, we locally perturb each replay sample in a targeted adversarial manner. The model then confuses them with the new tasks' data, creating more confusing examples close to the forgetting frontier as well as representative of previous data. The implementation code provided by the authors is at https://github.com/INK-USC/GMED.

## C   Reservoir Sampling

---

**Algorithm 6 Reservoir Update**

---

1: **Input:** Samples $(x_t, y_t)$, Buffer $\mathcal{M}$, reservoir/buffer size $m$
2: **if** $|\mathcal{M}| < m$ **then**
3:     $\mathcal{M} = \mathcal{M} \cup (x_t, y_t)$
4: **else**
5:     $j = \text{randint}(1, t)$
6:     **if** $j \leq m$ **then**
7:         $\mathcal{M}[j] \leftarrow (x_t, y_t)$
8:     **end if**
9: **end if**

---

We use traditional reservoir sampling method [62] detailed in Alg. 6 to update the limited memory buffer when a new batch's data comes in. Reservoir sampling uniformly samples from the incoming task's stream by assigning a sampling probability of $m/t$ where $t$ is the number of samples seen so far. It does not need any prior information about the length of the complete data stream.

Prior works based on task-free continual learning [2, 33, 37] have shown promising results when using reservoir sampling as the memory update method. [26, 37] have proposed modified versions of reservoir sampling which help populate the memory with diverse samples in a balanced way when dealing with heterogeneous and long-tailed data streams. Since, in this work, our focus is primarily on class-balanced datasets, we use Reservoir Sampling as the memory update method and plan to study other memory update methods which can be balanced in terms of the present classes in future work.

Table 4: Class-incremental Continual Learning Datasets Statistics

|  | Split-MNIST | Split-CIFAR10 | Split-CIFAR100 | Split-miniImageNet |
|---|---|---|---|---|
| # of tasks | 5 | 5 | 20 | 20 |
| # of classes per task | 2 | 2 | 5 | 5 |
| # of training samples per task | 1000 | 10000 | 2500 | 2500 |
| # of test samples per task | 2000 | 2000 | 500 | 500 |
| # of overall classes | 10 | 10 | 100 | 100 |

## D Dataset Details

Based on the taxonomy described in [61], the datasets that we studied fall under the *class-incremental* (CI) category. The dataset details are as follows:

**Split-MNIST** is a variant of the MNIST dataset (http://yann.lecun.com/exdb/mnist/) of handwritten digits [31] split into five disjoints tasks based on the labels. Each task consists of two classes, and we use only 1k examples from each task for training and report results on the complete test set. The goal is to classify all ten classes at the end of the last task.

**Split-CIFAR10** is a variant of the CIFAR-10 dataset [28] (https://www.cs.toronto.edu/ kriz/cifar.html) split into 5 disjoint tasks based on labels. Each task consists of two classes and 10k training examples.

**Split-CIFAR100** is a variant of the CIFAR-100 dataset (https://www.cs.toronto.edu/ kriz/cifar.html) comprises of 20 disjoint tasks with each task dealing with 5 classes. This is 100-way classification problem since we do not use any task information during training/testing.

**Split-miniImageNet** splits the mini-ImageNet dataset [14] (https://lyy.mpi-inf.mpg.de/mtl/download/) into 20 disjoint tasks based on labels. Each task comprises of 5 classes and 2.5k training examples.

In Table 4, we present the statistics of all four datasets. Training samples used for each task is $95\%$ of the mentioned numbers as $5\%$ is held-out as validation set. For fine-tuning the hyper-parameters associated with the RAR framework, we use the held-out validation set. In case of Split-MNIST dataset, following the setting of previous methods such as [2, 25], we use only 1000 training examples per task.

## E Complexity Analysis

We compare the replay-based methods in terms of forward and backward computations used, besides the forward features computed on the incoming batch $X_i$ and backward pass used for the final parameters updates:

1. **ER**: 1 extra forward pass on the replay samples i.e. $f(X'_M; \theta)$
2. **ER-RAR**: 1 backward pass for the virtual model update to $\theta'$, $n$ forward and backward passes to generate the targeted augmentations of $X'_M$ using $\theta'$ where $n$ denotes the number of iterative steps used to generate the RAR data, 1 forward pass on the replay examples and the generated RAR data using $\theta$.
3. **MIR**: 1 backward pass for the virtual model update to $\theta'$, 2 forward passes on the larger memory set, i.e. $f(X_M; \theta), f(X_M; \theta')$, 1 forward pass on the smaller set of replay samples i.e., $f(X'_M; \theta)$.
4. **MIR-RAR**: 1 backward pass for the virtual model update to $\theta'$, 2 forward passes on the larger memory set, i.e. $f(X_M; \theta), f(X_M; \theta')$, $n$ forward and backward passes to generate the targeted augmentations of $X'_M$ using $\theta'$ where $n$ denotes the number of iterative steps used to generate the RAR data, 1 forward pass on the smaller set of original replay samples and the RAR augmented data using $\theta$.

Thus, comparing ER-RAR w.r.t ER, ER-RAR needs $n$ extra forwards passes and $(n + 1)$ extra backward passes on top of ER. Similarly, MIR-RAR adds $n$ additional forward and backwards passes on top of MIR. Setting $n = 2$ for all datasets (except Split-MNIST with 1000 training samples per

task) meaning using only two gradient steps to optimize the RAR augmentation objective defined in Eq. 4 in the input space, we limit the additional computational costs introduced by RAR.

**Computational costs reduction based on Sec 3.2 and 3.3**: Instead of using all $|X_M|$ examples in the memory buffer $\mathcal{M}$, replay data selection reduces the potential candidates to be considered for adversarial data generation to $|X'_M|$. For example, given a buffer size of 500 and replay budget of $k = 10$, the potential buffer samples considered for generating RAR adversarial augmentations is only 10 instead of 500.

Having reduced the size of the replay set to $|X'_M|$, one would still need to consider every potential pairing between $X'_M$ and $X_i$ (current task samples) to find the most confusing subset of cardinality $c$ which incurs maximum RAR loss defined in Eq. 3. Here, $c$ is essentially the replay budget. Generating targeted adversarial samples for each pair can be computationally prohibitive. Our nearest neighbor-based pairing brings down the number of adversarial samples to be generated (by optimizing Eq. 3) to c from $|X'_M| \times |X_i|$. In the above example for a current batch size of 10, we only need to generate $c = 10$ targeted adversarial samples to approximately optimize Eq. 3 instead of generating $10 \times 10$ adversarial samples and then choosing $c$ samples with the largest RAR loss.

# F    Implementation Details

Following the current SoTA methods [2, 10, 25], we use a MLP classifier with two hidden layers as the backbone architecture for Split-MNIST dataset. For remaining datasets, we use the reduced ResNet-18 classifier [22] with the output layer having as many units as the number of classes present in the entire dataset. In all experiments based on experience replay methods, we use SGD optimizer with a learning rate of 0.1 for all datasets. For GMED [25], ASER [55] based methods, we present the results using the set of hyperparameters reported in the papers. Both mini-batch size and replay budget are set to 10 following above mentioned methods to make the comparisons fair. For MIR based methods, we set the subset size $c$ to 50 for all datasets following [2]. $c$ denotes the size of the larger set of buffer examples on which the forgetting/interference score is computed.

**Hyperparameters Tuning**: We tune the hyperparameters associated with the RAR framework $\{n, \beta, \epsilon\}$ on the held-out validation set across all datasets. For all datasets, we analyze (as shown in Figure 5) the sensitivity of the performance of RAR in terms of the number of gradient steps $n$ used to perform the constrained targeted optimization in the input space as shown in Eq. 4. Specifically, we study values of $n \in \{2, 5, 10\}$. Based on the trade-off between the performance and the computational costs associated with the forward and backward passes employed to generate the targeted adversarial augmentation, we set $n = 2$ for all datasets except for Split-MNIST where we set $n = 5$ based on the average forgetting score as shown in Fig. 5a. Since the MNIST dataset is relatively easier to train on, more gradient steps are needed to generate difficult samples which are indiscernible for the model in the learnt representation space after the look-ahead update on the new task's data.

For the weight assigned to the $\ell_{RAR}$ computed using Eq. 3 in the final loss objective (denoted by $\beta$), we search over $\{0.05, 0.075, 0.1, 0.2, 0.3, 0.4, 0.5\}$ and fine-tune them separately for different datasets across various buffer sizes that we study. For Split-MNIST, $\beta$ equal to $0.3, 0.3$, and $0.4$ lead to the best reported performance across buffer size $200, 500$, and $1000$ respectively. For Split-CIFAR10, these values are $0.5, 0.1$, and $0.075$ for buffer sizes $200, 500$, and $1000$ respectively. For Split-CIFAR100 and Split-miniImageNet, we get the best setting as $\beta = 0.4$.

We search for the best perturbation tolerance $\epsilon$ hyperparameter in $\{0.15, 0.3\}$ for Split-MNIST, $\{0.0314, 0.07\}$ for Split-CIFAR10 and Split-CIFAR100, and $\{0.01568, 0.0314\}$ for Split-miniImageNet. The best setting $\epsilon$ values are $0.3$ for Split-MNIST and $0.0314$ for remaining three datasets.

Prior works such as [46] show that the deeper/higher layers of neural networks are disproportionately responsible for catastrophic forgetting in line with the observation that the shallower/lower layers learn general features of and higher layers learn more task-specific features. Therefore, we use the penultimate layer (layer just before the final classification/softmax layer) as layer $l$ in Eq. 4 and in the representation space of $\theta'_l$, we minimize the augmentation objective such that the penultimate feature embedding of the RAR augmented sample and its paired current task sample are as close as possible, but visually still similar to the original buffer sample.

**Compute Resources**: We use a single NVIDIA GeForce GTX 1080 Ti to train our CL models. The PyTorch and CUDA toolkit versions used are 1.6.0 and 10.1 respectively.

# G    Limitations & Societal Impacts

**Limitations**: In this work, we have utilized Reservoir Sampling across all compared methods to update the memory/buffer as a new batch comes in. Using other recently proposed methods [3, 55] to tackle this problem could potentially help in maintaining a diverse & representative memory/buffer, particularly in case of class-imbalanced incremental datasets. We also plan to explore updating the memory/buffer with perturbed samples (in stead of original samples) and see if re-learning them at later stages of training could help ameliorate forgetting. Similar to other CL algorithms, our work is validated on small datasets with single pass training on streams generated in a synthetic fashion. Therefore, the final average accuracy achieved on all set of classes is significantly small compared to traditional (i.i.d. assumption based) supervised training on the entire dataset.

**Negative Societal Impacts**: The potential improvements brought in terms of mitigating forgetting while learning on a continual stream of data might have some indirect negative impact. For example, as spam detection systems are updated to prevent spam, bots can learn to continually evolve to defeat them. Similar issues could arise in other surveillance systems in place. The replay of RAR augmentations might introduce some bias if malicious attackers can manipulate the data stream in the continual learning setting.

# H    Background on Targeted Adversarial Augmentations (or Attacks) & Related Algorithms

Given a model $\theta$, associated layer $l$ of the model, source input-output pair $(x_s, y_s)$ and a target input-output pair $(x_t, y_t)$, the goal of targeted adversarial attack/perturbation is to generate an input $z$ which looks visually similar to the source input $x_s$, but in the latent space of model's $l$-th layer, the feature embedding of $z$ is close to the feature embedding of the target input $x_t$.

The objective can be mathematically summarized as:

$$g_{\theta_l}(x_s, x_t) \in \underset{z}{\mathrm{argmin}}\, \mathcal{L}_{\theta_l}(z, x_t) \tag{11}$$

$$\text{s.t. } ||z - x_s||_\infty \leq \epsilon \tag{12}$$

Here, if we use the model's last layer denoted by $L$, then often times $\mathcal{L}_{\theta_l}(z, x_t)$ can be replaced with the cross-entropy loss w.r.t. to the target label $y_t$ of the paired sample $x_t$ i.e., $\ell(z, y_t; \theta)$.

In case we use layer $l$ such that $l < L$, meaning layers lower/shallower than the last layer, one can use the defined distance function in Eq. 5 i.e., $d_{\theta_l}(z, x_t) := ||\theta_l(z) - \theta_l(x_t)||_2$ as the minimization objective $\mathcal{L}_{\theta_l}(z, x_t)$ in the above constrained optimization problem.

The Fast Gradient Sign Method (FGSM) proposed in [20] can be used to solve Eq. 11 when dealing with infinity norm constraints in the input space. Specifically:

$$g_{\theta_l}(x_s, x_t) = x_s - \epsilon \cdot \mathrm{sign}(\nabla_{x_s}\mathcal{L}_{\theta_l}(x_s, x_t)) \tag{13}$$

where $\nabla_{x_s}\mathcal{L}_{\theta_l}(x_s, x_t)$ is the gradient of the loss function in Eq. 11 w.r.t. $x_s$.

Iterative-FGSM [30] which iteratively applies FGSM $n$ times with a smaller step size $\alpha$ (proportional to $\epsilon/n$) can also be used. Here $n$ denotes the number of iterations used to iteratively optimize Eq. 11.

To stabilize the gradient update directions and avoid poor local maxima, [15] proposed Momentum-Iterative FGSM (MI-FGSM) which integrates momentum with a decay factor $\mu$ and accumulates the gradients of previous iterations while updating the targeted augmentation at each step.

At step $t = 0$, we initialize $x_s^0 = x_s$. Then, at step $t$, the accumulated gradient and targeted augmentation are derived as follows (the superscript $t$ denotes the iteration step, while the subscript $t$ denotes the target example):

$$g^{t+1} = \mu \cdot g^t + \frac{\nabla_{x_s}\mathcal{L}_{\theta_l}(x_s^t, x_t)}{||\nabla_{x_s}\mathcal{L}_{\theta_l}(x_s^t, x_t)||_1} \tag{14}$$

$$x_s^{t+1} = x_s^t - \alpha \cdot \mathrm{sign}(g^{t+1}) \tag{15}$$

At the end of each step, we project back into the norm ball $||x_s^{t+1} - x_s||_\infty \le \epsilon$ and we also clip the values back to the input range observed corresponding to the original pixel values when using image data. In this work, we use MI-FGSM with the decay factor $\mu$ as 1.0, the step-size $\alpha$ same as $\epsilon$ since we employ only two iterations to generate the RAR augmented data using buffer replay samples $(X_M', Y_M')$ as the source/anchor sample. The paired sample from the incoming batch of current task $(X_i, Y_i)$ can be seen as the target sample.

# I   Additional Experiments

Table 5: Comparison of RAR with other non-RAR baselines having similar computational costs. Numbers within bracket denote the overall buffer size used.

| Method | Split-MNIST (200) | | Split-CIFAR10 (200) | | Split-CIFAR10 (500) | | Split-CIFAR10 (1000) | |
| --- | --- | --- | --- | --- | --- | --- | --- | --- |
| | Accuracy ($\uparrow$) | Forgetting ($\downarrow$) | Accuracy | Forgetting | Accuracy | Forgetting | Accuracy | Forgetting |
| ER-3-iters | $82.3 \pm 0.8$ | $17.6 \pm 0.8$ | $28.5 \pm 1.3$ | $52.5 \pm 3.7$ | $36.1 \pm 1.5$ | $39.5 \pm 2.1$ | $41.5 \pm 1.2$ | $30.8 \pm 1.9$ |
| ER-GMED | $82.5 \pm 1.4$ | $15.0 \pm 1.7$ | $30.4 \pm 0.9$ | $39.0 \pm 1.3$ | $38.3 \pm 1.1$ | $27.7 \pm 1.3$ | $42.8 \pm 1.5$ | $20.5 \pm 1.2$ |
| ER-RAR (ours) | $\mathbf{84.0 \pm 1.1}$ | $\mathbf{11.8 \pm 1.5}$ | $\mathbf{33.2 \pm 1.4}$ | $\mathbf{37.6 \pm 2.1}$ | $\mathbf{40.4 \pm 1.7}$ | $\mathbf{25.1 \pm 2.5}$ | $\mathbf{44.0 \pm 1.4}$ | $\mathbf{18.1 \pm 2.0}$ |
| MIR-3-iters | $83.1 \pm 0.8$ | $15.3 \pm 1.4$ | $28.4 \pm 1.0$ | $53.4 \pm 2.1$ | $41.1 \pm 0.8$ | $33.3 \pm 1.5$ | $\mathbf{48.9 \pm 1.1}$ | $22.4 \pm 1.9$ |
| MIR-GMED | $84.2 \pm 1.2$ | $12.2 \pm 1.0$ | $27.5 \pm 1.3$ | $45.5 \pm 1.6$ | $37.2 \pm 1.1$ | $28.6 \pm 1.9$ | $43.0 \pm 1.5$ | $17.6 \pm 2.1$ |
| MIR-RAR (ours) | $\mathbf{86.2 \pm 0.6}$ | $\mathbf{9.1 \pm 0.8}$ | $\mathbf{33.3 \pm 1.1}$ | $\mathbf{33.5 \pm 2.2}$ | $\mathbf{43.1 \pm 1.1}$ | $\mathbf{21.1 \pm 2.2}$ | $45.3 \pm 1.5$ | $\mathbf{14.1 \pm 1.6}$ |

To ensure fair comparison in terms of computational costs (extra forward and backward passes as described in Sec. E), in Table 5, we compare RAR to GMED [25], ER-3-iters, and MIR-3-iters. 3-iters means that the model is trained for three iterations on the incoming batch and buffer samples as a new batch comes in. GMED also performs multiple iterations of gradient updates to edit/perturb the buffer samples. Across all compared methods, RAR-based methods perform significantly better in terms of forgetting (lower is better) as well as final accuracy.

**Comparison to knowledge distillation based methods**: On Split-CIFAR10 dataset for two different buffer sizes, we augment RAR with DER [6] which utilizes knowledge distillation in the logits space. DER which imposes regularization in the logits space can be seen as a complimnetary method to RAR and we observe that combining them together leads to further improvements in Table 6. ER-DER is essentially DER that uses random sampling to select the replay samples and was proposed in the DER paper. In MIR-DER, we use MIR scores to select the replay batch for DER. In Table 6, *aug* means that we apply standard augmentations such as random cropping and rotations on the replay samples.

Table 6: Comparison of RAR with DER (with and without augmentation) to non-RAR baselines. Numbers within bracket denote the overall buffer size used.

| Method | Split-CIFAR10 (200) | | Split-CIFAR10 (500) | |
| --- | --- | --- | --- | --- |
| | Accuracy ($\uparrow$) | Forgetting ($\downarrow$) | Accuracy ($\uparrow$) | Forgetting ($\downarrow$) |
| ER-DER | $28.0 \pm 1.3$ | $47.2 \pm 7.2$ | $32.9 \pm 1.9$ | $28.4 \pm 3.3$ |
| ER-RAR-DER | $36.0 \pm 1.6$ | $21.2 \pm 2.6$ | $39.8 \pm 0.5$ | $15.5 \pm 2.2$ |
| ER-DER-aug | $34.0 \pm 3.3$ | $37.4 \pm 7.6$ | $39.6 \pm 1.5$ | $18.2 \pm 4.1$ |
| ER-RAR-DER-aug | $36.7 \pm 1.5$ | $15.8 \pm 2.3$ | $40.3 \pm 1.1$ | $10.3 \pm 1.9$ |
| MIR-DER | $36.6 \pm 2.3$ | $20.0 \pm 1.5$ | $38.6 \pm 1.7$ | $10.8 \pm 1.9$ |
| MIR-RAR-DER | $35.9 \pm 1.9$ | $20.1 \pm 3.3$ | $38.5 \pm 0.8$ | $10.8 \pm 1.8$ |
| MIR-DER-aug | $36.5 \pm 2.3$ | $17.0 \pm 1.8$ | $39.5 \pm 1.0$ | $15.0 \pm 1.7$ |
| MIR-RAR-DER-aug | $37.2 \pm 1.8$ | $13.4 \pm 1.1$ | $40.1 \pm 0.8$ | $9.5 \pm 2.3$ |

**Comparison to other augmentation techniques**: In Table 3, we compare RAR to another perturbation method *random* which adds random noise of the same perturbation magnitude as RAR, and show that RAR leads to significant improvement in reducing forgetting as well as increasing the average accuracy on different CL benchmark datasets. In Table 7, we compare to other standard augmentations techniques such as random cropping and rotations applied to the replay samples. Utilizing these standard augmentation strategies along with RAR leads to impressive improvements in terms of the forgetting metric as shown in Table 7.

When using standard augmentations along with RAR, ER-RAR-aug and MIR-RAR-aug achieve similar performance irrespective of the memory retrieval/selection method used. This indicates that RAR, when combined with standard augmentations such as random cropping and rotations, is more

Table 7: Comparison of RAR to standard augmentation techniques. Numbers within bracket denote the overall buffer size used.

| Method | Split-CIFAR10 (200) | | Split-CIFAR10 (500) | |
| --- | --- | --- | --- | --- |
| | Accuracy ($\uparrow$) | Forgetting ($\downarrow$) | Accuracy ($\uparrow$) | Forgetting ($\downarrow$) |
| ER | $25.7 \pm 1.3$ | $51.3 \pm 4.1$ | $31.9 \pm 0.9$ | $39.0 \pm 2.1$ |
| ER-aug | $30.0 \pm 4.1$ | $43.2 \pm 6.9$ | $39.4 \pm 2.8$ | $26.0 \pm 6.3$ |
| ER-RAR | $33.2 \pm 1.4$ | $37.6 \pm 2.1$ | $40.4 \pm 1.7$ | $25.1 \pm 2.5$ |
| ER-RAR-aug | $39.3 \pm 1.6$ | $21.6 \pm 3.4$ | $42.2 \pm 1.9$ | $13.6 \pm 2.5$ |
| MIR | $28.0 \pm 1.5$ | $48.1 \pm 2.9$ | $37.5 \pm 1.6$ | $32.2 \pm 2.1$ |
| MIR-aug | $36.1 \pm 2.6$ | $23.7 \pm 3.6$ | $40.5 \pm 1.2$ | $18.4 \pm 3.3$ |
| MIR-RAR | $33.3 \pm 1.1$ | $33.5 \pm 2.2$ | $43.1 \pm 1.1$ | $21.1 \pm 2.2$ |
| MIR-RAR-aug | $41.6 \pm 1.2$ | $21.0 \pm 2.8$ | $42.2 \pm 2.3$ | $12.5 \pm 2.6$ |

robust to replay data selection and can generate high-quality replay samples (diverse, confusing, and representative of previously seen tasks).

However, such augmentation strategies rely on strong human priors, and manually selecting such augmentation operators requires domain expertise and prior knowledge of the dataset of interest [70]. They may perform poorly in the scenarios when human knowledge is weak for the targeted domain. For example, for medical image analysis, such simple transformations are shown to be insufficient in capturing many of the subtle variations present in the data [69]. Moreover, they are static and pre-defined before the training so they cannot capture the training dynamics during the continual learning process for the purpose of reducing the forgetting. RAR, on the other hand, automatically generates augmentations adaptive to the forgetting dynamics of the model and thus focuses on reducing the local interference near the forgetting frontier between different tasks.

**Virtual update for RAR samples generation**: When training sequentially on a set of tasks, catastrophic forgetting of previous tasks occurs as the network weights become biased to meet the objectives of the new task. Performing a single-step virtual update of the model by exposing it only to the current task's data helps us explore which buffer samples are most likely to be forgotten. We use $\theta'$, the virtually updated model to generate the adversarial samples using Eq. 4 as such perturbed samples are likely to be forgotten in future updates and confused with the current task's classes leading to increased loss on them. In retrospect, using the generated perturbed sample (visually close to the original sample $x_M$ in the input space and hence in-distribution) with its correct label $y(x_M)$ and replaying it while we are still at $\theta$ helps in alleviating forgetting. As we combine the losses, we may also consider the procedure as first doing an update on the incoming task to get $\theta'$ and then performing an adversarial training step based on $\theta'$ using the RAR augmented samples.

One can argue that the step size used to perform this virtual update ($\eta_{\text{virtual}}$) could be different from the step size ($\eta$) used to update the continual learner such that the adversarial samples generated using $\theta'$ are not too easy or too hard to classify by the current model $\theta$. We perform experiments on the Split-CIFAR10 dataset with a buffer size of 500 and an SGD optimizer with a learning rate ($\eta$) of 0.1. We find that using the same learning rate as the SGD optimizer to perform the virtual update results in the best performance on MIR-RAR as shown in Table 8.

Table 8: Performance of MIR-RAR for different values of $\eta_{\text{virtual}}$ when $\eta = 0.1$

| $\eta_{\text{virtual}}$ | Accuracy ($\uparrow$) | Forgetting ($\downarrow$) |
| --- | --- | --- |
| No virtual update | $41.7 \pm 1.3$ | $21.8 \pm 1.4$ |
| 0.02 | $42.1 \pm 1.6$ | $22.1 \pm 2.8$ |
| 0.05 | $42.6 \pm 1.1$ | $20.6 \pm 3.7$ |
| 0.1 | $43.1 \pm 1.1$ | $21.1 \pm 2.2$ |
| 0.15 | $43.1 \pm 1.6$ | $22.1 \pm 3.5$ |
| 0.2 | $41.9 \pm 1.1$ | $23.1 \pm 3.6$ |

**Ratio between incoming mini-batch size and replay budget**: Following prior works [2, 25, 55], we select the same number of replay samples from the buffer as the mini-batch size of the incoming new task. This is set to 10 irrespective of the buffer size. Here, we perform additional experiments where we vary the ratio between the incoming mini-batch size and replay budget. As expected, when

the combined batch is dominated primarily by the incoming task samples (mini-batch size / replay budget = 15/5), we observe a significant drop on the overall performance with respect to both the average accuracy and forgetting as shown in Table 9. However, when replaying more samples from the buffer (mini-batch size / replay budget = 5/15) in each training iteration, which over-compensates for forgetting, we do not observe any improvement in reducing the forgetting. This suggests that to achieve a good plasticity-stability trade-off, the ratio between incoming mini-batch size and replay budget need to be close to one.

Table 9: Performance of replay-based CL methods when using different ratios between incoming mini-batch size and replay budget. Dataset used is Split-CIFAR10 with a buffer size of 500.

| Ratio | 10/10 | | 5/15 | | 15/5 | |
|---|---|---|---|---|---|---|
| Method | Accuracy ($\uparrow$) | Forgetting ($\downarrow$) | Accuracy ($\uparrow$) | Forgetting ($\downarrow$) | Accuracy ($\uparrow$) | Forgetting ($\downarrow$) |
| ER | $31.9 \pm 0.9$ | $39.0 \pm 2.1$ | $30.5 \pm 2.7$ | $53.0 \pm 4.4$ | $20.9 \pm 2.1$ | $55.7 \pm 6.1$ |
| ER-RAR | $40.4 \pm 1.7$ | $25.1 \pm 2.5$ | $40.2 \pm 2.2$ | $26.0 \pm 3.2$ | $26.0 \pm 3.6$ | $31.4 \pm 3.5$ |
| MIR | $37.5 \pm 1.6$ | $32.2 \pm 2.1$ | $37.4 \pm 1.4$ | $35.1 \pm 2.0$ | $16.5 \pm 1.5$ | $68.5 \pm 3.3$ |
| MIR-RAR | $43.1 \pm 1.1$ | $21.1 \pm 2.2$ | $41.0 \pm 1.4$ | $22.9 \pm 3.2$ | $18.6 \pm 2.6$ | $58.0 \pm 5.1$ |

# J Visualizations of Learnt Decision Regions

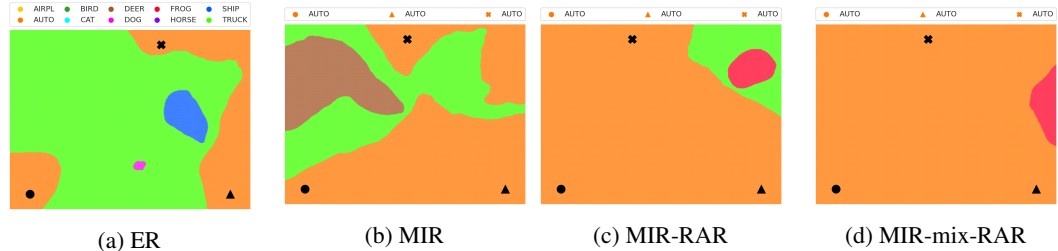

    (a) ER          (b) MIR         (c) MIR-RAR      (d) MIR-mix-RAR

Figure 7: Decision regions are generated using a triplet of training samples on which the model makes the right predictions. All samples are from same class *Auto (Task 1)*. Legend in (a) applies to all and describes the color used to denote different classes. Remaining legends denote the classes corresponding to the chosen triplet samples.

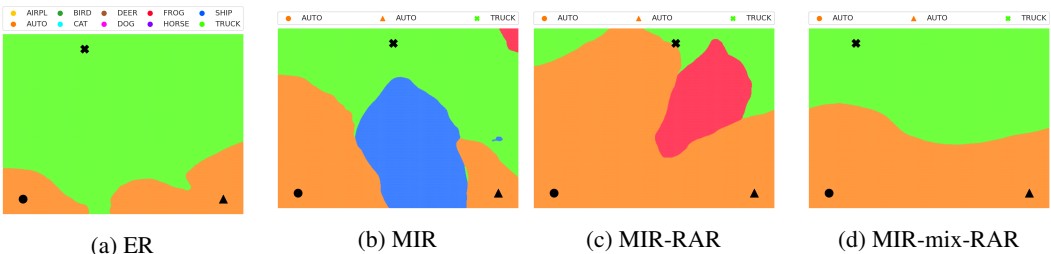

    (a) ER          (b) MIR         (c) MIR-RAR      (d) MIR-mix-RAR

Figure 8: Decision regions are generated using a triplet of training samples on which the model makes the right predictions. Two samples belong to class *Auto (Task 1)*, while the third samples is from class *Truck (Task 5)*. Legend in (a) applies to all and describes the color used to denote different classes. Remaining legends denote the classes corresponding to the chosen triplet samples.

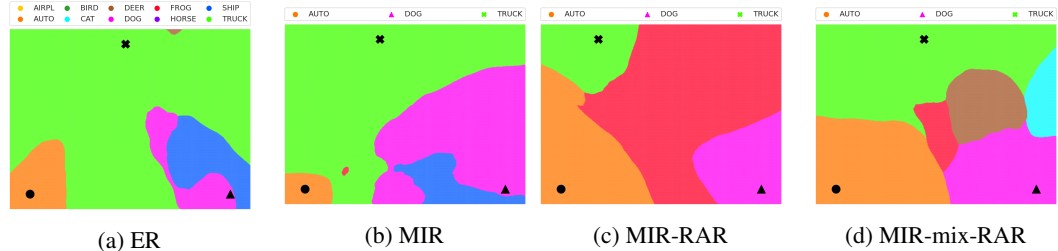

| (a) ER | (b) MIR | (c) MIR-RAR | (d) MIR-mix-RAR |

Figure 9: Decision regions are generated using a triplet of training samples on which the model makes the right predictions. Samples are from the following classes: *Auto (Task 1), Dog (Task 3), Truck (Task 5)*. Legend in (a) applies to all and describes the color used to denote different classes. Remaining legends denote the classes corresponding to the chosen triplet samples.

Inspired from recent works in decision boundaries visualization [57], we try to visualize the decision regions learnt by the continual learner along the data manifold. To plot the decision regions, we use the plotting technique from [57] (https://github.com/somepago/dbViz). Using a sampled triplet of data points $(x_1, x_2, x_3)$, we construct a plane spanned by vectors $\overrightarrow{v_1} = x_2 - x_1$ and $\overrightarrow{v_2} = x_3 - x_1$ and plot decision regions in this plane.

Specifically, for our use-case, once the training on all tasks (5 in case of Split-CIFAR10) is finished, we use the trained model to make predictions on the trained data. We, then, select three training set instances $(x_1, x_2, x_3)$ on which the model makes the correct predictions in following ways: (1) all samples are from the same class, (2) two samples share the same class label while the third belongs to a different class, and (3) all three samples are from different classes.

We analyze the decision regions learnt by the ResNet-18 model using ER [10], MIR [2], MIR-RAR, and MIR-mix-RAR methods on the Split-CIFAR10 dataset. Task to classes mapping is as follows: **Task 1: {AIRPL, AUTO}, Task 2: {BIRD, CAT}, Task 3: {DEER, DOG}, Task 4: {FROG, HORSE}, Task 5: {SHIP, TRUCK}**.

For case (1), we select all samples from class *Auto* and plot the regions in Fig 7. It is expected that in case of worse forgetting, the decision regions would be dominated by the last task classes even when we use the first task's samples to model the visualization plane. In Fig. 7, the plane is mostly dominated by green (and blue) color in case of ER. In case of MIR-mix-RAR, we see the plane mostly covering the class used to construct the visualization plane, even when the class selected (*Auto*) belongs to the first task the model encountered.

Similarly, in Fig. 8 (case (2)) & 9 (case (3)), as we go from left to right, we see the decision regions being more structured around the classes covered in the triplet set and less dominated by the color of the classes from the last task — this demonstrates that forgetting is minimized when using RAR and mix-RAR.