# OpenReview forum: "Retrospective Adversarial Replay for Continual Learning"
_NeurIPS.cc/2022/Conference — NeurIPS 2022 Accept_

### Official Review · Reviewer_GiLn · 2022-06-30

**Rating:** 6
**Confidence:** 3
**Soundness:** 2 fair
**Presentation:** 3 good
**Contribution:** 3 good

**Summary:**

Authors propose a method to mitigate Catastrophic Forgetting named “Retrospective Adversarial Replay (RAR)”, which synthesizes adversarial samples near the forgetting boundary. The method perturbs a buffered sample towards its nearest neighbour drawn from the current task in latent representation space. In particular, the authors propose Retrospective Aversarial replay Loss that finds a pairing between memory and novel task samples such that memory samples that are perturbed towards the paired current-task samples is maximum. By replaying the perturbed sample with the original label authors prohibit novel tasks boundaries to interfere with the previous tasks boundaries. By these means, they refine the boundary between previous and current tasks and reduce bias towards the current task. In addition, the authors propose also a mix-RAR strategy that utilizes MixUp augmentation on memory samples that are later perturbed using RAR. The proposed modification improves the class boundaries refinement and reduces the final average forgetting.

The main contribution of the papers is an introduction of a novel interesting strategy for data retrieval from the replay buffer that is based on the combination of past and present data. The authors evaluate the method on top of other buffer-retrival-related methods and show consistent performance gain in terms of averaged accuracy on a number of datasets. Additionally, the method outperforms another buffer samples perturbation method called Gradient-based Memory EDiting Examples (GMED) and random perturbation baseline.

**Questions:**

1) Was another kind of replay samples augmentation also tested such as cropping/noise injection/random rotations?
2) Did this kind of augmentation provide noticeable improvements?

**Ethics Review Area:**

["I don’t know"]

**Limitations:**

The work does not exhibit potential negative societal impact. The authors are aware of the limitations of their work and address them accordingly.


**Strengths And Weaknesses:**

Originality:
	The proposed method introduces a novel improvement for the memory buffer methods with the inspiration drawn from psychological phenomena called “false memories”. The method differs from the other memory perturbation buffer replay method GMED to which it directly compares by calculating differently the perturbation direction. Specifically, in RAR memory examples perturbation is in the direction of the novel tasks' decision boundaries. In contrast, in GMED authors edit examples so that they are more likely to be forgotten in future updates.

Quality:
	The experimental results clearly support the effectiveness of the method. RAR outperforms random perturbation of the memory examples and GMED in the scenario where RAR, random perturbation and the memory examples are used with Experience Replay (ER) and Maximally Interfered Retrieval (MIR). Additionally, RAR on top of other memory-retrieval methods such as ER, GSS, ASER, and MIR results in improved averaged accuracy and reduces forgetting. The method’s effectiveness is shown on four datasets with different resolutions.

Clarity:
	The method and its effectiveness are plainly explained with an intuitive illustration, brief formulas, and results section that provides a decent comparison with other methods. The overall structure of the paper is well organized and the comparative study is appropriately conducted.

Significance:
	The authors address the problems with ER, namely memory records selection and their perturbation in a clever way, considering the decision boundaries of the tasks that are known to degenerate in continual learning. Even though this kind of decision boundaries analysis is not new it is the first application of this technique in the memory replay methods.

Strengths:
- The proposed method invariably outperforms other methods based on the perturbation of buffer samples in terms of forgetting mitigation.
- The method is deeply grounded in our current understanding of catastrophic forgetting i.e. that the decision boundaries in classifier models deteriorate on every new task.
- Authors propose also a MixUp-based strategy to increase replay variation
- Authors provide extensive ablations about their method to prove the correctness of it at different steps, such as Figure 2, for the step of distance calculation between samples from the buffer and from the novel tasks.

Weaknesses:
- Authors could provide quantities of how their improvements from sections 3.2 and 3.3 reduce the computational cost.

---

> ### Author Response · Authors · 2022-08-02
> **Initial Response to Reviewer GiLn**
>
> Thank you for your detailed and constructive review, as well as your positive feedback on the writing clarity, significance, and novelty of our approach. We address your comments and pertinent questions about using augmentations below.
>
> **Response to the weakness concern:**
>
> Section 3.2: Instead of using all $|X_M|$ examples in the memory buffer $\mathcal{M}$, replay data selection reduces the potential candidates to be considered for adversarial data generation to $|X’_M|$ [Lines 176-177]. For example, given a buffer size of 500 and replay batch size of 10, the potential buffer samples considered for generating RAR adversarial replay samples is only 10 instead of 500.
>
> Section 3.3: Having reduced the size of the replay set to $|X’_M|$, one would still need to consider every potential pairing between $X’_M$ and $X_i$ (current task samples) to find the most confusing subset of cardinality $c$ which incurs maximum RAR loss defined in Eq. 3. Generating targeted adversarial samples for each pair can be computationally prohibitive [Lines 168-174]. Our nearest neighbor-based pairing brings down the number of adversarial samples to be generated (by optimizing Eq. 3) to c from $|X’_M| \times |X_i|$. In the above example for a current batch size of 10, we only need to generate $c=10$ targeted adversarial samples to approximately optimize Eq. 3 instead of generating $10 \times 10$ adversarial samples and then choosing $c$ samples with the largest RAR loss.
>
>
> **Response to questions 1 & 2:**
>
> In Table 3 of the main paper, we compare RAR to another perturbation method “random” which adds random noise of the same perturbation magnitude as RAR, and show that RAR leads to significant improvement in reducing forgetting as well as increasing  the average accuracy on different CL benchmark datasets. Regarding standard augmentations techniques such as random cropping and rotations, we report results on the Split-CIFAR10 for two different buffer sizes: 200 and 500. Utilizing these standard augmentation strategies leads to impressive improvements in terms of the forgetting metric as shown in the table below.
>
> When using standard augmentations along with RAR, ER-RAR-aug and MIR-RAR-aug achieve similar performance irrespective of the memory retrieval/selection method used. This indicates that RAR, when combined with standard augmentations such as random cropping and rotations, is more robust to replay data selection and can generate high-quality replay samples (diverse, confusing, representative of previously seen tasks).
>
> |  | Mem = 200 |  | Mem = 500 |  |
> |---|---|---|---|---|
> | Method | Accuracy  | Forgetting | Accuracy  | Forgetting |
> | ER | 25.7 +/- 1.3 | 51.3 +/- 4.1 | 31.9 +/- 0.9 | 39.0 +/- 2.1 |
> | ER-aug | 30.0 +/- 4.1 | 43.2 +/- 6.9 | 39.4 +/- 2.8 | 26.0 +/- 6.3 |
> | ER-RAR | 33.2 +/- 1.4 | 37.6 +/- 2.1 | 40.4 +/- 1.7 | 25.1 +/- 2.5 |
> | ER-RAR-aug | 39.3 +/- 1.6 | 21.6 +/- 3.4 | 42.2 +/- 1.9 | 13.6 +/- 2.5 |
> |  |  |  |  |  |
> | MIR | 28.0 +/- 1.5 | 48.1 +/- 2.9 | 37.5 +/- 1.6 | 32.2 +/- 2.1 |
> | MIR-aug | 36.1 +/- 2.6 | 23.7 +/- 3.6 | 40.5 +/- 1.2 | 18.4 +/- 3.3 |
> | MIR-RAR | 33.3 +/- 1.1 | 33.5 +/- 2.2 | 43.1 +/- 1.1 | 21.1 +/- 2.2 |
> | MIR-RAR-aug | 41.6 +/- 1.2 | 21.0 +/- 2.8 | 42.2 +/- 2.3 | 12.5 +/- 2.6 |
>
> However, such augmentation strategies rely on strong human priors, and manually selecting such augmentation operators requires domain expertise and prior knowledge of the dataset of interest (reference [1] below). However, they may perform poorly in the scenarios when human knowledge is weak for the targeted domain. For example, for medical image analysis, such simple transformations are shown to be insufficient in capturing many of the subtle variations present in the data (reference [2] below).
>
> Moreover, they are static and pre-defined before the training so they cannot capture the training dynamics during the continual learning process for the purpose of reducing the forgetting. RAR, on the other hand, automatically generates augmentations adaptive to the forgetting dynamics of the model and thus focuses on reducing the local interference near the forgetting frontier between different tasks.
>
> References:
>
> [1] Deep Autoaugment, Zheng et al., 2022
>
> [2] Data augmentation using learned transformations for one-shot medical image segmentation, Zhao et al., 2019

---

> > ### Author Response · Authors · 2022-08-08
> > **Looking forward to discuss our paper and revisions with you**
> >
> > We thank you sincerely for your thorough and helpful review! **In order to address your concerns about the computational cost reduction and the comparison to standard data augmentation methods, we have reported new experimental results along with detailed explanations in the response.** We hope that our responses have addressed your concerns and would hugely appreciate the opportunity to further incorporate your suggestions to improve our work if you get a chance to read our response.

---

> > > ### Comment · Reviewer_GiLn · 2022-08-08
> > > **Thank you**
> > >
> > > Thank you for addressing my comments and questions. I sustain my score and weakly recommend acceptance of this work.

---

### Official Review · Reviewer_KXx6 · 2022-07-05

**Rating:** 6
**Confidence:** 4
**Soundness:** 3 good
**Presentation:** 3 good
**Contribution:** 3 good

**Summary:**

The paper proposes a novel approach to tackle the classifier bias problem in the context of continual learning. For this purpose, the approach (called RAR) focusses on the examplars that are close to the the forgetting border. RAR perturbs the selected exemplars towards the closest sample from the current task in the latent space. By replaying such samples, RAR is able to refine the boundary between
previous and current tasks, hence combating forgetting and reducing bias towards the current task. Moreover, the authors propose to combine RAR with the mix-up technique which significantly improves continual learning in the small buffer regime. RAR is a generic approach and could be combined with any experience replay method.

**Questions:**

Here are my concerns:
- Equation 2: It should be y(X'_M) instead of y(Y'_M)
- When you perturb a sample from the X'_M what label do you assign to it? The label of the closest sample in X_i?
- Section 3.3: I found it confusing that you refer to x_M as a sample from X'_M. Why do not use the x'_M notation instead? I think it will improve the clarity. Same observation for Algorithm 1
- Algorithm 1: 'S' should be also mentioned in the input
- Do you maintain a fixed size for the buffer? Are the previous classes uniformly represented in the buffer?
- What is the ratio in the mini-batch between the exemplars and the samples belonging to the new task? I think performing a study of different ratios would be interesting.

**Limitations:**

Yes

**Strengths And Weaknesses:**

Strengths:
- the paper is in general clearly written
- the approach is well-motivated and scientifically sound
- the review of the state of the art covers most of the related works
- experimental results are extensive and demonstrate the superiority of the proposed approach
- a solid analysis of the proposed approach is provided (sensitivity analysis of the hyper-parameters, ablation studies, etc.)

Weaknesses:
- some aspects needs to be clarified
- some mathematical notation produces confusion

---

> ### Author Response · Authors · 2022-08-02
> **Initial Response to Reviewer KXx6**
>
> Thank you for your review as well as your positive feedback on the writing clarity, scientific soundness, and extensive empirical validation of our proposed approach. We address your questions below:
>
> **Response to the questions:**
> 1) Thank you for noticing this typo. It should be $y(X’_M)$.
>
> 2) For perturbing a sample $x_M \in  X’_M$, we pair it with another sample from the current task $x_i \in X_i$ and then perform the targeted adversarial perturbation of RAR. The perturbed sample (after optimizing Eq. 4) is similar to the original sample $x_M$ (visually indistinguishable on their images) but its latent embedding is close to the latent embedding of the paired current task sample $x_i$ [Lines 155-162]. When we train the model on such perturbed samples, we utilize the correct label of $x_M$  which is the original replay sample as shown in Eq. 3.
>
> 3) Thank you for the suggestion. We will change it to $x’_M$ in the main paper.
>
> 4) In Algorithm 1, we initialize $S$ as an empty set (line 7) and then populate it with the pairings as described in Section 3.3.
>
> 5) Yes, in our experiments, we maintain a fixed-size buffer. We tested different buffer sizes for datasets such as Split-CIFAR10, Split-MNIST, and Split-CIFAR100 and observed that RAR performs more effectively on mitigating forgetting than other methods, especially in a small buffer regime as shown in Fig. 4b.
>
> 	We use reservoir sampling to update the memory buffer which uniformly samples from the incoming stream [Lines 131-133]. Thus, in principle, all seen classes should be uniformly represented in the buffer. However, in the case of long-tailed datasets, several previous works (references [1] below and [26] & [37] from the main paper) have studied class-balancing reservoir sampling to maintain a representative and sizable set of under-represented classes in the memory. As mentioned in lines 65-67, RAR being generic in nature can be easily integrated with any memory update methods (orthogonal to our work).
>
> 	References:
>
> 	[1] Online Continual Learning from Imbalanced Data, Chrysakis, et al., 2020
>
> 6) Following prior works in the continual learning literature such as [2, 25, 55], we select the same number of replay samples from the buffer as the mini-batch size of the incoming new task. This is set to 10 irrespective of the buffer size as mentioned in lines 238-239.
> Following your suggestion, we performed new experiments on the Split-CIFAR10 dataset with a buffer size of 500. As expected, when the combined batch is dominated primarily by the incoming task samples (minibatch size / replay size = 15/5), we observe a significant drop on the overall performance with respect to both the average accuracy and forgetting. However, when replaying more samples from the buffer (minibatch size / replay size = 5/15) in each training iteration, which over-compensates for forgetting, we do not observe any improvement in reducing the forgetting. This suggests that to achieve a good plasticity-stability trade-off, the ratio between replay samples and incoming batch samples need to be close to one. We will report more experiments comparing different ratios on other CL datasets in the next revision.
>
> 	|  | Accuracy (mem = 500) |  |  |
> 	|---|---|---|---|
> 	| **Method** | **10/10** | **5/15** | **15/5** |
> 	| ER | 31.9 +/- 0.9 | 30.5 +/- 2.7 | 20.9 +/- 2.1 |
> 	| ER-RAR | 40.4 +/- 1.7 | 40.2 +/- 2.2 | 26.0 +/- 3.6 |
> 	| MIR | 37.5 +/- 1.6 | 37.4 +/- 1.4 | 16.5 +/- 1.5 |
> 	| MIR-RAR | 43.1 +/- 1.1 | 41.0 +/- 1.4 | 18.6 +/- 2.6 |
>
> 	|  | Forgetting (mem = 500) |  |  |
> 	|---|---|---|---|
> 	| **Method** | **10/10** | **5/15** | **15/5** |
> 	| ER | 39.0 +/- 2.1 | 53.0 +/- 4.4 | 55.7 +/- 6.1 |
> 	| ER-RAR | 25.1 +/- 2.5 | 26.0 +/- 3.2 | 31.4 +/- 3.5 |
> 	| MIR | 32.2 +/- 2.1 | 35.1 +/- 2.0 | 68.5 +/- 3.3 |
> 	| MIR-RAR | 21.1 +/- 2.2 | 22.9 +/- 3.2 | 58.0 +/- 5.1 |

---

> > ### Comment · Reviewer_KXx6 · 2022-08-06
> > **Response to Authors**
> >
> > I would like to thank the authors for addressing my concerns stated in the initial review. I will take them in consideration when making the final decision.

---

### Official Review · Reviewer_K76z · 2022-07-10

**Rating:** 5
**Confidence:** 5
**Soundness:** 2 fair
**Presentation:** 3 good
**Contribution:** 3 good

**Summary:**

This paper aims at overcoming catastrophic forgetting in continual learning. Catastrophic forgetting is the main challenge in continual learning. This paper propose a method called Retrospective Adversarial Replay (RAR) to address the catastrophic forgetting.

**Questions:**

(a) The proposed method seems to be effective in the experiments. But I'm not sure that the proposed method would also work in theory. I try to summarize my arguments. If i made some mistakes, please point them out and give some reasonable explanations. (1) The data of different tasks should follow different distributions, which often makes the data of different tasks far away from each other. Therefore, the permuted data may not lead to a great improvement. (2) In my view, the permuted samples are close to current task's samples in the latent space. Therefore, the model is not trained with the samples which replay previous tasks' knowledge---adversarial samples perform similarly to the current task's samples in the latent space.

(b) Why do authors use look-ahead parameters $\theta^{\prime}$ instead of the current parameters $\theta$ to generate the adversarial samples? To my knowledge, the generation of adversarial samples depends on the model's parameters. Therefore, the adversarial samples generated by the model with look-ahead parameters $\theta^{\prime}$ can not equal to the adversarial samples which should be generated by the model with $\theta$. Please provide a reasonable explanation.

**Limitations:**

This work does not seem to have a great negative societal impact.

**Strengths And Weaknesses:**

Strengths:

(a) This paper tries to use adversarial samples to defy the catastrophic forgetting in continual learning. It is novel.

(b) This paper provides extensive experiments and the proposed method seems to be effective.

(c) This paper is well-written and easy to read.

Weaknesses:

(a) I am concerned about the idea of this paper. I summarize my views in Questions. If authors provide reasonable and clear explanations, This paper is worth to being accepted.

---

> ### Author Response · Authors · 2022-08-02
> **Initial Response to Reviewer K76z**
>
> Thank you for your review and helpful feedback on the writing clarity, novelty, and empirical strength of our proposed approach. Regarding your questions which we find to be insightful, please see our detailed responses below.
>
> **Response to the questions:**
>
> 1) In different continual learning scenarios such as class incremental, domain incremental, etc., it is indeed the case that input distributions vary across different tasks. Having a fixed small memory buffer and repeatedly replaying the same samples stored in the limited buffer leads to the model overfitting on the buffered samples and thus degenerate the generalization performance [Lines 36-40].
>
> 	[Lines 155-162] The adversarially perturbed sample generated by optimizing Eq. (4) is close to the original replay sample $x_M$ in the input space. This is ensured by the $\ell_\infty$ norm-based constraint in the input space. As a result of the minimization objective in Eq. (4), in the latent space of the model, the perturbed samples’ feature embeddings are more close to the current samples’ feature embeddings and the model ends up confusing the perturbed sample as the current samples’ class. Training the model on such perturbed samples closer to the decision boundary as well as representative of the previous tasks helps the model retain the knowledge of previous tasks. Also, note that in the RAR loss (Eq. 3), we provide the correct label $y(x_M)$ for the perturbed sample. This helps the model in correcting the wrong predictions (e.g. the current task’s classes $y(x_i)$).
>
> 	In Figure 3 for the Split-MNSIT dataset, we provide some examples of the RAR-generated data augmentations that look visually similar to their original buffer samples but in the latent space, each perturbed sample’s embedding is closer to its paired sample from the current task. When the model is trained on the pair of perturbed sample and the label of the original buffer sample $y(x_M)$, it is able to correct its prediction from the current task’s class to the right class $y(x_M)$.
>
> 2) When training sequentially on a set of tasks, catastrophic forgetting of previous tasks occurs as the network weights become biased to meet the objectives of the new task [Lines 27-30][27]. **Performing a single-step virtual update of the model by exposing it only to the current task’s data helps us explore which buffer samples are most likely to be forgotten.** We use $\theta’$, the virtually updated model to generate the adversarial samples using Eq. 4 as such perturbed samples are likely to be forgotten in future updates and confused with the current task’s classes leading to increased loss on them. In retrospect, using the generated perturbed sample (close to the original sample $x_M$ and hence in-distribution) with their correct label $y(x_M)$ and replaying it while we are still at $\theta$ helps in alleviating forgetting [Lines 162-167]. As we combine the losses, we may also consider the procedure as first doing an update on the incoming task to get $\theta’$ and then performing an adversarial training step based on $\theta’$ using the RAR augmented samples.
>
> 	One can argue that the step size used to perform this virtual update could be different from the step size (learning rate) used to update the continual learner such that the adversarial samples generated using $\theta’$ are not too easy or too hard to classify by the current model $\theta$. We perform experiments on the Split-CIFAR10 dataset with a buffer size of 500 and an SGD optimizer with a learning rate (lr) of 0.1. We find that **using the same learning rate in the optimizer to perform the virtual update result in the best performance on MIR-RAR**. We will add more extensive experiments in the updated draft of the supplementary materials.
>
> 	|  | Accuracy | Forgetting |
> 	|---|---|---|
> 	| no virtual update | 41.7 +/- 1.3 | 21.8 +/- 1.4 |
> 	| lr 0.02 | 42.1 +/- 1.6 | 22.1 +/- 2.8 |
> 	| lr 0.05 | 42.6 +/- 1.1 | 20.6 +/- 3.7 |
> 	| lr 0.1 | 43.1 +/- 1.1 | 21.1 +/- 2.2 |
> 	| lr 0.15 | 43.1 +/- 1.6 | 22.1 +/- 3.5 |
> 	| lr 0.2 | 41.9 +/- 1.1 | 23.1 +/- 3.6 |

---

> > ### Author Response · Authors · 2022-08-08
> > **Looking forward to discuss our paper and revisions with you**
> >
> > We thank you sincerely for your thorough and helpful review! **In order to address your question about using look-ahead update and concerns about using perturbed data, we have added new experimental results along with detailed explanations in the response.** We hope that our responses have addressed your concerns and would hugely appreciate the opportunity to further incorporate your suggestions to improve our work if you get a chance to read our response.

---

### Official Review · Reviewer_f9rr · 2022-07-11

**Rating:** 4
**Confidence:** 4
**Soundness:** 2 fair
**Presentation:** 3 good
**Contribution:** 2 fair

**Summary:**

This paper presents a method called Retrospective Adversarial Replay (RAR) to synthesize adversarial examples from the buffered data that are close to the current data in the latent space. Thus, the decision boundary between the previous and current task got refined to mitigate forgetting. The authors further propose MixUp-based variant for the small-buffer regime. Experiments on class-incremental benchmarks demonstrate the effectiveness of the proposed method.

**Questions:**

1. As far as I understand, the proposed method could be treated as a data-augmentation method on the buffered data. Although it improves upon a certain sets of replay methods in table 1 and 2, and several data perturbation methods in table 3, I wonder how it performs agains some other state-of-the-art replay based methods, such as DER [1], Co$^2$L [2], etc.

2. The motivation of only evaluating the proposed method under the class-incremental scenario is not that clear. What about other CL settings such as domain-incremental learning?

It is important to see if the proposed method as a general replay-based method that outperforms prior state-of-the-arts, or a more incremental adds-on to a limited category of methods.

[1] Buzzega, Pietro, et al. "Dark experience for general continual learning: a strong, simple baseline." Advances in neural information processing systems 33 (2020): 15920-15930.
[2] Cha, Hyuntak, Jaeho Lee, and Jinwoo Shin. "Co2l: Contrastive continual learning." Proceedings of the IEEE/CVF International Conference on Computer Vision. 2021.

**Limitations:**

I did not find a clear section (or pointers to) describe the limitations of the work. The potential negative societal impacts are not discussed in the paper.

**Strengths And Weaknesses:**

Strengths:
- The paper is well-written and easy to follow.
- Introducing adversarial examples in CL is interesting.
- The proposed method is easy to be combined with retrieval-based replay methods.

Weakness:
- Strength 3 is actually a weakness considering the generalizability of the method. It seems that stage 1 depends a lot on the retrieval-based strategies, if the selected samples are not close to the boundary, then the following stages will not further generate effective adversarial examples.
- The experiments part seems insufficient. See questions for more details.

---

> ### Author Response · Authors · 2022-08-02
> **Initial Response to Reviewer f9rr**
>
> Thank you for your review and positive feedback on the writing clarity and simplicity of our framework. We found your questions really valuable and address them below.
>
> **Responses to the questions:**
>
> 1) We perform new experiments of DER which utilizes knowledge distillation in the logits space on the Split-CIFAR10 dataset for two different buffer sizes: 200 and 500. ER-DER is essentially DER that uses random sampling to select the replay samples and was proposed in the DER paper. In MIR-DER, we use MIR scores to select the replay batch for DER. “aug” means that we apply standard augmentations such as random cropping and rotations on the replay samples.
>
> 	**We observe consistent improvements brought by RAR when using different memory retrieval methods in DER**: (1) When using ER (random replay), RAR-based methods outperform their non-RAR counterparts quite significantly in terms of forgetting along with improvements in average accuracy; (2) When using MIR and standard augmentations, we see similar improvements.
>
> 	| Method | Mem = 200 |  | Mem = 500 |  |
> 	|---|---|---|---|---|
> 	|  | Accuracy | Forgetting | Accuracy | Forgetting |
> 	| ER-DER | 28.0 +/- 1.3 | 47.2 +/- 7.2 | 32.9 +/- 1.9 | 28.4 +/- 3.3 |
> 	| ER-RAR-DER | 36.0 +/- 1.6 | 21.2 +/- 2.6 | 39.8 +/- 0.5 | 15.5 +/- 2.2 |
> 	|  |  |  |  |  |
> 	| ER-DER-aug | 34.0 +/- 3.3 | 37.4 +/- 7.6 | 39.6 +/- 1.5 | 18.2 +/- 4.1 |
> 	| ER-RAR-DER-aug | 36.7 +/- 1.5 | 15.8 +/- 2.3 | 40.3 +/- 1.1 | 10.3 +/- 1.9 |
> 	|  |  |  |  |  |
> 	| MIR-DER | 36.6 +/- 2.3 | 20.0 +/- 1.5 | 38.6 +/- 1.7 | 10.8 +/- 1.9 |
> 	| MIR-RAR-DER | 35.9 +/- 1.9 | 20.1 +/- 3.3 | 38.5 +/- 0.8 | 10.8 +/- 1.8 |
> 	|  |  |  |  |  |
> 	| MIR-DER-aug | 36.5 +/- 2.3 | 17.0 +/- 1.8 | 39.5 +/- 1.0 | 15.0 +/- 1.7 |
> 	| MIR-RAR-DER-aug | 37.2 +/- 1.8 | 13.4 +/- 1.1 | 40.1 +/- 0.8 | 9.5 +/- 2.3 |
>
> 	In our experiments on all datasets, we use a single epoch training setting meaning only a single pass through a task’s data is considered. This setting is more realistic in practical scenarios as well as in line with the assumption that the memory buffer is limited in size. For these reasons, we could not directly compare to Co2L which uses multiple epochs settings for each task’s data. However, we believe that the RAR perturbed samples that represent the memory buffer (previous tasks) can serve as hard negatives in the supervised contrastive loss formulation used in the Co2L paper.
>
> 2) Following the terminology from [60], we empirically validate our proposed framework on different class-incremental datasets which focus on incrementally learning new classes of objects. In the domain-incremental scenario, where the input distribution changes across tasks while all the tasks share the same set of classes, RAR-generated replay samples can help the model distinguish between different domains by generating confusing samples anchored around previously seen domains.
>
> 	Following the settings in [2], we present results on the Permuted-MNIST dataset which is generated by applying 10 different permutations to the original MNIST dataset to generate 10 different tasks. Similar to [2], we train using only 1000 samples for each task. **Both ER-RAR and MIR-RAR methods outperform their non-RAR counterparts in terms of average accuracy and forgetting for different buffer sizes.**
>
> 	|  | Mem = 200 |  | Mem = 500 |  | Mem = 1000 |  |
> 	|---|---|---|---|---|---|---|
> 	| Method | Accuracy  | Forgetting | Accuracy  | Forgetting | Accuracy  | Forgetting |
> 	| ER | 75.9 +/- 0.5 | 6.4 +/- 0.5 | 78.9 +/- 0.4 | 3.8 +/- 0.3 | 81.0 +/- 0.3 | 2.3 +/- 0.3 |
> 	| ER-RAR | 79.1 +/- 0.5 | 4.6 +/- 0.5 | 82.5 +/- 0.4 | 2.2 +/- 0.3 | 84.7 +/- 0.2 | 0.9 +/- 0.1 |
> 	|  |  |  |  |  |  |  |
> 	| MIR | 75.7 +/- 0.6 | 6.9 +/- 0.5 | 79.4 +/- 0.5 | 4.3 +/- 0.4 | 81.9 +/- 0.2 | 2.2 +/- 0.3 |
> 	| MIR-RAR | 78.9 +/- 0.5 | 4.9 +/- 0.6 | 82.7 +/- 0.4 | 2.5 +/- 0.3 | 84.9 +/- 0.4 | 1.2 +/- 0.2 |

---

> > ### Author Response · Authors · 2022-08-02
> > **Contd. Response**
> >
> > **Response to the weakness concern about dependence on memory retrieval methods:**
> >
> > In RAR, stage I first identifies buffer samples for replay. Based on the selected samples, RAR generates perturbed samples (close to the original replay samples) that are most likely to be confused with the current task samples via pairing (Sec 3.3) and optimizing Eq. 4 [Lines 155-162].
> >
> > The improvement brought by RAR does not necessarily depend on the memory retrieval methods: it improves both the methods retrieving replay samples closer to the decision boundary (e.g., MIR & ASER) and ER (experience replay), which randomly selects replay samples from the buffer. Tables 1 & 2 show how ER-RAR and ER-mix-RAR outperform ER and ER-mix respectively. For small buffer sizes such as 200, the improvement in average forgetting is very significant on Split-CIFAR10. Also, for datasets such as Split-CIFAR100 and Split-miniImagenet comprising 20 tasks overall, the improvement brought by using RAR in terms of accuracy and forgetting is non-trivial even when the memory retrieval method used is ER. Since RAR captures the local interference between different tasks, it is more robust to the nature of the memory retrieval method used and, in totality, helps mitigate forgetting.
> >
> > **RAR**: Our proposed RAR framework is generic and complimentary to many existing CL methods. It can be easily integrated into existing frameworks for CL such as regularization methods  (DER, Co2L, etc.) as well as different memory retrieval & update methods (ER, MIR, ASER, GEM, and others). RAR’s ability to capture forgetting at local boundaries between the previously seen tasks and incoming tasks in the form of adversarially perturbed replay samples helps in mitigating catastrophic forgetting and improving generalization on all tasks.
> >
> > **Limitations**: We have addressed the limitations of our proposed framework in Appendix F (lines 756-764).
> >
> > **Negative Societal Impacts**: The potential improvements brought in terms of mitigating forgetting while learning on a continual stream of data might have some indirect negative impact. For example, as spam detection systems are updated to prevent spam, bots can learn to continually evolve to defeat them. Similar issues could arise in other surveillance systems in place. The replay of RAR augmentations might introduce some bias if malicious attackers can manipulate the data stream in the continual learning setting.

---

> > > ### Author Response · Authors · 2022-08-08
> > > **Looking forward to discuss our paper and revisions with you**
> > >
> > > We thank you sincerely for your thorough and helpful review! **In order to address your concerns about the genericity of RAR and its relevance to the domain-incremental continual learning scenario, we have added new experimental results along with detailed explanations in the response.** We hope that our responses have addressed your concerns and would hugely appreciate the opportunity to further incorporate your suggestions to improve our work if you get a chance to read our response.

---

### Meta-Review · Area_Chair_Q7y5 · 2022-08-31

**Recommendation:** Accept
**Confidence:** Less certain

**Metareview:**

This paper proposes adversarially perturbed samples from a replay buffer to simulate examples that are on "forgetting boundary" in the continual learning setting. After reading the paper, I found it interesting and insightful: the methodology makes sense, the analysis of their method is thorough, and the results validate their method. The reviewers asked some good questions, and I believe the authors did a good job at answering them sufficiently.

I therefore recommend the paper for acceptance. We ask however that the authors fix first line in the abstract which seems to suggest that all CL methods use a memory buffer.

I'm disappointed the reviewers did not participate more: they were reminded both my me as well as the authors about the discussion, yet no one participated after they gave their initial, albeit useful, signal.

**Award:**

No

---

### Decision · Program_Chairs · 2022-09-14

Accept